# Host-genotype dependent gut microbiota drives zooplankton tolerance to toxic cyanobacteria

Emilie Macke[1], Martijn Callens[1], Luc De Meester[2] & Ellen Decaestecker[1]

The gut microbiota impacts many aspects of its host's biology, and is increasingly considered as a key factor mediating performance of host individuals in continuously changing environments. Here we use gut microbiota transplants to show that both host genotype and gut microbiota mediate tolerance to toxic cyanobacteria in the freshwater crustacean *Daphnia magna*. Interclonal variation in tolerance to cyanobacteria disappears when *Daphnia* are made germ-free and inoculated with an identical microbial inoculum. Instead, variation in tolerance among recipient *Daphnia* mirrors that of the microbiota donors. Metagenetic analyses point to host genotype and external microbial source as important determinants of gut microbiota assembly, and reveal strong differences in gut microbiota composition between tolerant and susceptible genotypes. Together, these results show that both environmentally and host genotype-induced variations in gut microbiota structure mediate *Daphnia* tolerance to toxic cyanobacteria, pointing to the gut microbiota as a driver of adaptation and acclimatization to cyanobacterial harmful algal blooms in zooplankton.

[1] Laboratory of Aquatic Biology, Department of Biology, University of Leuven—Campus Kulak, E. Sabbelaan 53, B-8500 Kortrijk, Belgium. [2] Laboratory of Aquatic Ecology, Evolution and Conservation, University of Leuven, Charles Deberiotstraat 32, 3000 Leuven, Belgium. Correspondence and requests for materials should be addressed to E.M. (email: emilie.macke@kuleuven.be) or to E.D. (email: ellen.decaestecker@kuleuven.be)

In recent years, evidence has accumulated that the gut microbiota is not just a random set of microorganisms, but rather a complex community that plays a critical role in host physiology[1]. In particular, gut symbionts provide their host with crucial metabolic capabilities, such as digestion of plant polysaccharides[2] and metabolism of xenobiotic bioactive molecules like diet-derived toxins[3] or human-crafted poison[4]. Recently, processes once thought to depend solely on the host's genome, such as immunity[5], energy storage,[6] and thermal homeostasis[7], were shown to be strongly influenced by the gut microbiota. All these processes are likely involved in providing means of survival when facing environmental stress such as pollution, eutrophication, or atmospheric warming[8]. Hence, the gut microbiota also increasingly appears as a key factor driving the maintenance of fitness in the face of human-induced environmental change.

Unlike intracellular symbionts, which are strictly vertically transmitted to the embryo, gut-associated microbes are mainly acquired during and after birth via horizontal transfer or recruitment of bacteria from the surrounding environment[9–11]. Hence, they represent extra genetic and functional diversity, and provide the host with a broad set of metabolic functions[1,9]. Contrary to the host genome, which is largely static, the microbiome is highly plastic, and can respond rapidly to changes in host diet or environmental conditions, through changes in community composition, mutations, exchange of genetic material with bacteria from the environment, or changes in gene expression[1,8,9]. The gut microbiota is thus an important source of metabolic flexibility for the host, and might be a key, yet understudied, factor driving fast acclimatization to new environments, also in the current context of fast and drastic environmental changes imposed by human activities and climate warming[8,12]. In addition to fostering phenotypic plasticity, the gut microbiota is increasingly hypothesized to contribute to host evolution and genetic adaptation to the environment. Under particular environmental conditions, individuals harboring a set of symbionts that provides them with fitness advantages will be more successful, and this can influence evolutionary trajectories, including selection for host genotypes that are associated with specific metagenomes[8].

Several premises need to be fulfilled if the microbiota is to increase the host's acclimatization or adaptation capacity, but empirical studies so far did not study these simultaneously[1,8]. First, the characteristics of the microbiota must impact host phenotype and provide a fitness benefit to the host under particular environmental conditions. So far, several studies have shown the existence of variation in gut microbiota composition within host species depending on diet[2] or on host genetic background[13], but very few studies have measured the impact of such variation on host phenotype and fitness[6,7]. Most studies rely on correlative data and speculate about possible effects. There is thus a need to investigate phenotypic variation after transplant of gut microbiota acclimated to specific environmental conditions, or originating from different host genotypes, combined with the measurement of survival and fitness after microbiota transplants[1,8]. Second, if the microbiota is to influence evolutionary processes, nuclear and microbial genes need to be co-inherited for selection to operate on their interactions[14]. It is thus crucial to decipher how gut microbial communities are assembled. Shapira[9] recently defined the gut microbiota as a multilayered structure, composed of both a flexible pool of microbes that depends on the diversity of the environmental microbial pool and external conditions, and a core microbiota of host-specific microbes that are selectively recruited depending on host genetic factors, most likely mediated by the immune system. Supporting the existence of such core microbiota, recent studies revealed that host genetic background contributes to shape variation in gut microbiota

composition within a range of host species, including insects, fish, mice, and humans[1]. The relative contribution of external (e.g., exogenous microbial exposure) vs. internal (e.g., host genotype) factors to gut microbiota composition, however, remains poorly known.

Predicting how, and to what extent, the gut microbiota may drive acclimatization and adaptation thus requires integrative studies investigating the factors responsible for variation in gut microbiota structure, but also the consequences of such variations for host fitness[8,15]. The freshwater crustacean *Daphnia magna* offers unique opportunities for such studies. Its high experimental tractability, short life cycle, clonal reproduction, and high responsiveness to environmental stressors, combined with the possibility to easily manipulate its gut microbiota, provide a unique opportunity to study the interplay between host genotype, environment, and microbiota, with a high degree of experimental control[16,17].

Here we used the *D. magna* system to investigate the role of gut microbes in their host's response to an environmental stressor. Specifically, we tested the hypothesis that the gut microbiota mediates *Daphnia* tolerance to toxic cyanobacteria, which are responsible for increasingly problematic harmful blooms in ponds and reservoirs worldwide due to eutrophication and climate warming (i.e., cyanobacterial harmful algal blooms, or cyanoHABs)[18]. Because they release powerful toxins, cyanoHABs pose severe threats to livestock and human health, causing diseases ranging from gastrointestinal symptoms to liver cancer[19]. In aquatic ecosystems, cyanoHABs have a strong negative impact on zooplankton grazers, and through the food web, disrupt the whole freshwater community[20]. Deciphering the mechanisms underlying resistance to toxic cyanobacteria in these grazers is essential to predict how cyanoHABs can be prevented or controlled[20]. In *Daphnia*, previous studies have reported genetic variation in resistance to toxic cyanobacteria[21–23], and both genetic adaptation[21,22] and acclimatization[24] have been reported in *Daphnia* populations exposed to cyanoHABs[21]. Combining gut microbiota transplants with a metagenetic approach, we here show that both host genotype- and environmentally induced variations in the gut microbiota mediate *Daphnia* tolerance to toxic cyanobacteria, pointing to the gut microbiota as a potential important driver of adaptation and acclimatization to cyanoHABs in this key grazer. Monitoring changes in gut microbial community composition over the *Daphnia* life cycle following a transplant revealed that the taxonomic composition is mainly determined by exogenous microbial exposure in early life stages, and then progressively diverge among genotypes, indicating a selective recruitment of bacteria depending on host genetic background.

## Results

**The gut microbiota mediates *Daphnia* tolerance to cyanobacteria.** In the absence of manipulation of the microbiota, there were pronounced differences in tolerance to the toxic cyanobacteria *Microcystis aeruginosa* (strain PCC 7806 wild type, that produces microcystin) among *Daphnia* genotypes (genotype × diet interaction: $\chi_3^2 = 14.58$, $p = 0.002$, Cox proportional hazard model, hereafter called Cox model). Two genotypes (*S1* and *S2*) were found to be susceptible as they showed a significant decrease in survival when fed the toxic cyanobacteria, compared to non-toxic food (i.e., the green algae *Scenedesmus obliquus*; Fig. 1; Supplementary Fig. 1a, b). Two other genotypes (*T1* and *T2*) were more tolerant, as their survival was either increased or non-affected, respectively, when fed toxic cyanobacteria (Fig. 1; Supplementary Fig. 1c, d). Overall, upon cyanobacterial exposure, the two tolerant genotypes exhibited similar survival patterns ($\chi_1^2 = 1.24$, $p = 0.26$, pairwise comparison in Cox model), and

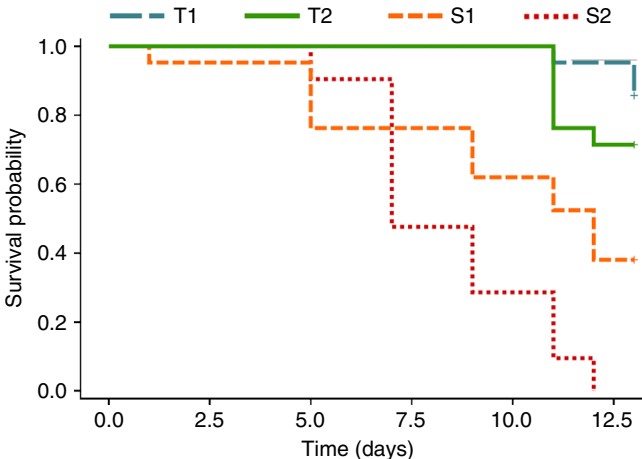

**Fig. 1** Survival in the four *Daphnia* genotypes used in the transplant experiments following cyanobacterial exposure. Colored lines indicate the different genotypes (tolerant genotypes: *T1* and *T2*, in blue and green, respectively; susceptible genotypes: *S1* and *S2*, in orange and red, respectively). The two tolerant genotypes (*T1* and *T2*) exhibited similar survival patterns ($p = 0.26$), and survived longer than the two susceptible genotypes (*S1* and *S2*; $p = 0.002$, $p < 0.0001$, $p = 0.02$ and $p < 0.0001$ for *T1–S1*, *T1–S2*, *T2–S1*, and *T2–S2* pairwise comparisons, respectively). Among susceptible genotypes, *S1* had a higher survival rate than *S2* ($p = 0.001$). $p$ values were obtained from a Cox model. Sample size was $n = 21$ (7 individuals × 3 biological replicates) for each genotype

survived longer than the two susceptible genotypes ($\chi_1^2 = 9.2$, $p = 0.002$; $\chi_1^2 = 24.6$, $p < 0.0001$; $\chi_1^2 = 5.5$, $p = 0.02$; and $\chi_1^2 = 24.1$, $p < 0.0001$ for *T1–S1*; *T1–S2*; *T2–S1* and *T2–S2* pairwise comparisons in Cox model, respectively; Fig. 1). Among the susceptible genotypes, *S1* had a higher survival rate than *S2* when fed toxic food ($\chi_1^2 = 10.3$, $p = 0.001$; Fig. 1).

To determine the role of gut microbiota in *Daphnia* tolerance to cyanobacteria, reciprocal gut microbiota transplants were performed among *Daphnia* genotypes and diet (Fig. 2). Donor *Daphnia* from tolerant (*T1* and *T2*) and susceptible (*S1* and *S2*) genotypes were provided with a same inoculum of pond water to ensure a sufficiently diverse gut microbiota, and exposed to either a nontoxic green algal diet or a toxic cyanobacterial diet for 12 generations. These exposures were performed in 2 L glass jars containing populations of ~60 individuals, with 3 independent replicates for each genotype × diet combination (i.e., 4 genotypes × 2 diets × 3 replicates = 24 donor populations). The gut microbiota from these donor *Daphnia* was then extracted, and inoculated into germ-free recipient juveniles from either tolerant or susceptible genotypes. We evaluated the tolerance of recipient *Daphnia* by exposing them to a cyanobacterial diet for two weeks (Fig. 2).

The results of the transplant experiment revealed that when *Daphnia* were made germ-free, and inoculated with similar gut microbiota, all genotypes exhibited similar levels of tolerance to toxic cyanobacteria. Indeed, overall, all recipient genotypes had similar survival and reproduction probability upon cyanobacterial exposure ($\chi_2^2 = 0.90$, $p = 0.64$ and $\chi_2^2 = 3.81$, $p = 0.15$, respectively; Cox model and logistic regression, respectively; Fig. 3a, d). The genetic variation in tolerance to cyanobacteria disappeared when reciprocal gut microbiota transplants were performed in the recipient genotypes, suggesting that gut microbiota rather than *Daphnia* genotype itself drives tolerance to toxic cyanobacteria.

While tolerance to toxic cyanobacteria did not depend on recipient genotype, it was strongly affected by donor genotype

(survival: $\chi_3^2 = 109.84$, $p < 0.0001$, Cox model, Fig. 3b; reproduction: $\chi_3^2 = 79.73$, $p < 0.0001$, logistic regression, Fig. 3e). *Daphnia* inoculated with gut microbiota from susceptible donor genotypes had a lower survival than *Daphnia* inoculated with gut microbiota from tolerant genotypes ($\chi_1^2 = 41.2$, $p < 0.0001$; $\chi_1^2 = 136.9$, $p < 0.0001$; $\chi_1^2 = 54.8$, $p < 0.0001$; and $\chi_1^2 = 126.4$, $p < 0.0001$ for *T1–S1*; *T1–S2*; *T2–S1* and *T2–S2* pairwise comparisons in Cox model, respectively; Fig. 3b). In addition, *Daphnia* inoculated with the gut microbiota of the *S1* genotype (i.e., the least susceptible of the susceptible genotypes) had a higher survival than *Daphnia* inoculated with the gut microbiota of the *S2* genotype (i.e., the most susceptible genotype; $\chi_1^2 = 41.2$, $p < 0.0001$). Upon cyanobacterial exposure, the survival pattern observed in recipient *Daphnia* thus largely reflected that of their donor genotypes (Fig. 1). Similar results were obtained for reproduction, with a higher probability of reproduction for *Daphnia* inoculated with the gut microbiota of tolerant donors ($\chi_1^2 = 24.9$, $p < 0.0001$; $\chi_1^2 = 49.1$, $p < 0.0001$; $\chi_1^2 = 30.9$, $p < 0.0001$; $\chi_1^2 = 54.4$, $p < 0.0001$; and $\chi_1^2 = 3.6$, $p = 0.06$ for *T1–S1*; *T1–S2*; *T2–S1*; *T2–S2*; and *S1–S2* contrasts in logistic regression, respectively; Fig. 3e). Hence, while protection against cyanobacteria can be transferred through the gut microbiota, the strength of the protection depends on the donor genotype. Together, these results demonstrate that genetic variation in *Daphnia* tolerance to toxic cyanobacteria is mediated by genotype-dependent gut microbiota.

Our results indicate that a pre-exposure of donors to *Microcystis* significantly increased the protective effect of the microbiota against cyanobacteria, fostering higher survival ($\chi_1^2 = 18.43$, $p < 0.0001$, Cox model, Fig. 3c) and probability of reproduction ($\chi_1^2 = 27.84$, $p < 0.0001$, logistic regression, Fig. 3f) in recipient *Daphnia*. This suggests that gut microbiota responded to become more efficient in dealing with toxic cyanobacteria after prior exposure. The strength of pre-exposure benefits, however, varied among donor genotypes (survival: $\chi_3^2 = 27.00$, $p < 0.0001$, Cox model; reproduction: $\chi_3^2 = 10.00$, $p = 0.019$, logistic regression; Supplementary Fig. 2).

**Gut microbiota composition in tolerant and susceptible genotypes**. To compare the gut microbiota of resistant and susceptible genotypes, and to determine the effects of a cyanobacterial exposure on the microbiota structure, the gut microbiota composition of the 24 donor populations described above was characterized. The *Daphnia* gut microbiota was extracted (using a pool of 20 guts per population) and sequenced in the V4 region of bacterial 16S rRNA, resulting in 901,179 high-quality reads and an average of 37,549 reads/sample. The results showed a strong effect of *Daphnia* genotype on the taxonomic composition of the gut microbiota (Fig. 4a–c). Especially, a clear distinction could be made between the gut microbiota of tolerant genotypes, dominated by Flavobacteria ($78 \pm 19.4\%$ and $80 \pm 18.8$ in green algal and cyanobacterial diet, respectively), and that of susceptible genotypes, dominated by Betaproteobacteria ($79 \pm 8.4\%$ and $81 \pm 10.6$ in green algal and cyanobacterial diet, respectively). The quantification of β-diversity, based on the weighted UniFrac distance, revealed that most of the variation in the gut microbiota composition was explained by this tolerant vs. susceptible genotype effect ($p = 0.001$, permutation multivariate analysis of variance (MANOVA); Fig. 4c). Diet, however, did not have a significant impact on the taxonomic composition of the gut microbiota ($p = 0.287$, permutation MANOVA; Fig. 4a, c).

Whereas α-diversity was not affected by diet ($p = 0.16$ and 0.62 for species richness and the Shannon index, respectively, analysis of variance (ANOVA); Fig. 4b), it significantly differed between tolerant and susceptible genotypes, with a higher diversity in

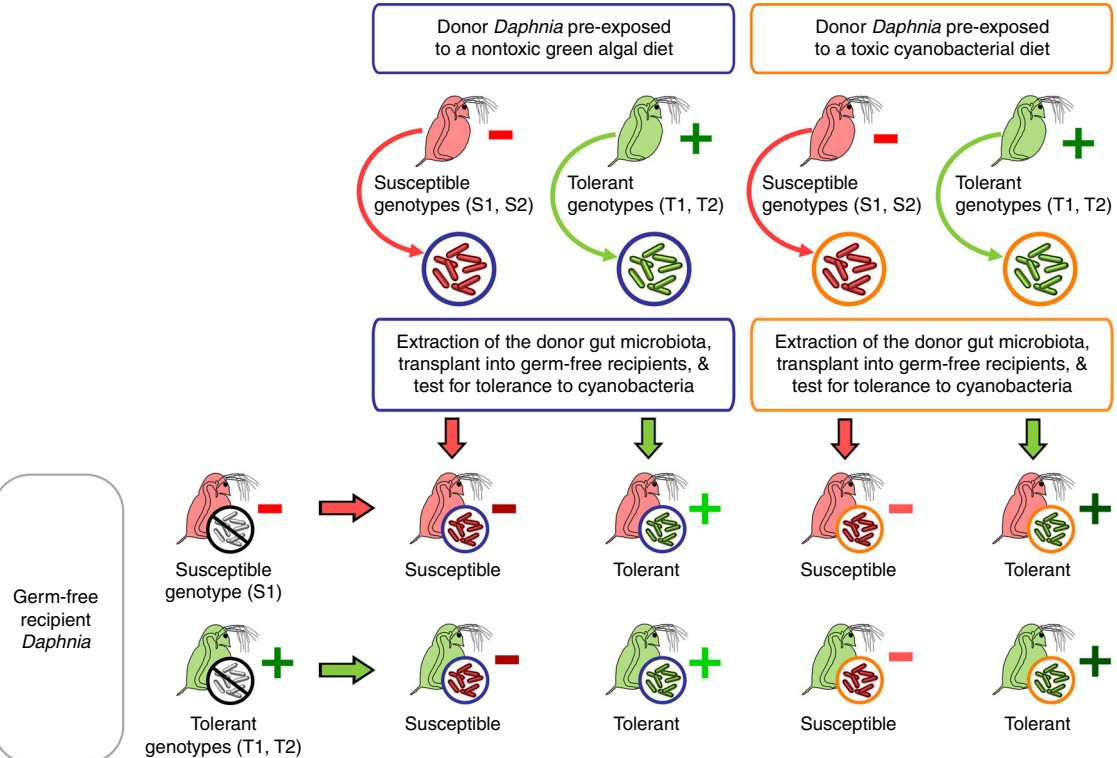

**Fig. 2** Deciphering how genotype and gut microbiota interact to drive tolerance to toxic cyanobacteria through gut microbiota transplants in *Daphnia*. Donor *Daphnia* populations from either susceptible (*S1* and *S2*; red) or tolerant (*T1* and *T2*; green) genotypes were exposed to either a nontoxic green algal or a toxic cyanobacterial diet (blue and orange color, respectively) for 12 generations. Their gut microbiota was then extracted by crushing dissected guts, and inoculated into germ-free recipient *Daphnia* juveniles from susceptible (*S1*; red) and tolerant (*T1* and *T2*; green) genotypes. Recipient *Daphnia* were subsequently fed on a cyanobacterial diet to determine their tolerance. We found that the level of tolerance to cyanobacteria in recipient *Daphnia* was co-determined by donor genotype and donor diet, and was not influenced by recipient genotype. Symbols "+" and "−" indicate the performance (i.e., survival and reproduction) of recipient *Daphnia* upon cyanobacterial exposure, from the lowest performance (dark red "−") to the highest (dark green "+"). Green and red colors indicate tolerance and susceptibility, respectively

susceptible genotypes ($p = 0.0006$ and $0.0002$ for species richness and the Shannon index, respectively, ANOVA; Fig. 4b). Although Betaproteobacteria were dominant in susceptible genotypes, other classes such as Sphingobacteria and Gammaproteobacteria were also present, and more represented than in tolerant genotypes ($p = 0.012$ and $0.023$, respectively, Wald test on log2-fold change estimates, hereafter called "Wald test"). In addition, for a given class of bacteria, the gut microbiota of susceptible genotypes exhibited more variation at the family level than that of tolerant genotypes (Supplementary Fig. 3a).

To analyze more precisely the structural basis of the observed pattern, we investigated what OTUs (at the family level) exhibited the most dramatic difference between susceptible and tolerant genotypes, using the log2-fold change estimate. The strongest difference was observed in Flavobacteriaceae, which were strongly under-represented in susceptible compared to tolerant genotypes (0.9% and 79%, respectively; $p < 0.0001$, Wald test; Supplementary Fig. 3b). On the other hand, Comamonadaceae, Neisseriaceae, Microbacteriaceae, and Saprospiraceae were overrepresented in susceptible genotypes (36%, 25%, 4% and 3%, respectively, in susceptible genotypes, compared to 8%, 6%, 1% and 0.1% in resistant genotypes; $p < 0.001$ for all comparisons, Wald test; Supplementary Fig. 3b).

**Assembly and temporal dynamics of the *Daphnia* gut microbiota**. To investigate the dynamics of the gut microbial community after a gut microbiota transplant, and to determine the relative contribution of colonization (i.e., exogenous microbial

exposure) vs. internal sorting (i.e., genotype) processes to the gut microbiota structure, we performed an additional experiment in which we exposed germ-free *Daphnia* genotypes to different microbial inocula, and monitored their gut microbiota composition over time.

Germ-free *Daphnia* juveniles of two distinct genotypes (hereafter called "*Recipient 1*" and "*Recipient 2*", which were different from *T1*, *T2*, *S1*, and *S2* genotypes described above) were inoculated with four different bacterial inocula (A, B, C, and D) obtained by extracting the gut microbiota from four distinct *Daphnia* populations (3 replicates per type of "inoculum × recipient genotype" combination, with 30 *Daphnia* in each replicate; see Methods for further details). The gut microbiota composition in recipient *Daphnia* was then monitored at different time points after the transplant (1, 7, and 14 days), by extracting and pooling the gut microbial communities of 10 *Daphnia* per experimental unit. The V4 region of bacterial 16S rRNA was sequenced using MiSeq technology, resulting in 3,838,246 high-quality reads and an average of 51,176 reads/sample.

The microbiota successfully established in the digestive tract of *Daphnia* juveniles, as bacteria were detected in the gut already 1 day after the transplant (Fig. 5a). Overall, the composition of the microbiota that initially established in the gut was different from that of the inocula ($p = 0.004$, permutation MANOVA; Fig. 5a, b). For example, Betaproteobacteria were abundant in *Daphnia* that received inoculum C, while the inoculum itself was dominated by Gammaproteobacteria (Fig. 5a). In *Daphnia* that received the inocula A and B, the opposite effect was observed.

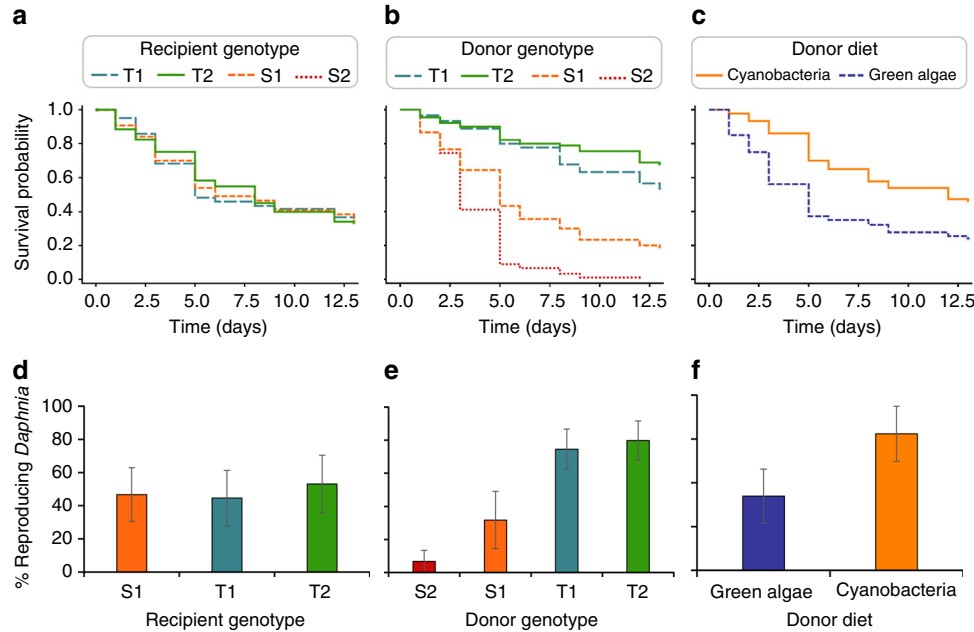

**Fig. 3** Survival and reproduction upon cyanobacterial exposure in recipient *Daphnia* that received a gut microbiota transplant. **a** Average survival of recipient *Daphnia*, grouped by recipient genotype. The effect of recipient genotype on recipient *Daphnia* survival was not statistically significant ($p = 0.64$). **b** Average survival of recipient *Daphnia*, grouped by donor genotype. Survival in recipient *Daphnia* was significantly affected by the donor genotype ($p < 0.0001$), with higher survival rates in recipient *Daphnia* receiving the gut microbiota from tolerant donor genotypes ($p = 0.02$ for *T1–T2* pairwise comparison and $p < 0.0001$ for all other pairwise comparisons). **c** Average survival of recipient *Daphnia*, grouped by donor diet. A pre-exposure of donors to toxic cyanobacteria significantly increased survival in recipients, compared to a pre-exposure to green algae ($p < 0.0001$). **d** Average proportion of reproducing *Daphnia*, grouped by recipient genotype. The effect of recipient genotype on *Daphnia* reproduction was not statistically significant ($p = 0.15$). **e** Average proportion of reproducing *Daphnia*, grouped by donor genotype. The proportion of reproducing recipient *Daphnia* was significantly affected by the donor genotype ($p < 0.0001$), and was higher in recipient *Daphnia* receiving the gut microbiota from tolerant donor genotypes ($p < 0.0001$ for *T1–S1*, *T1–S2*, *T2–S1*, and *T2–S2* pairwise comparisons; $p = 0.058$ and $0.14$ for *S1–S2* and *T1–T2* pairwise comparisons, respectively). **f** Average proportion of reproducing *Daphnia*, grouped by donor diet. A pre-exposure of donors to toxic cyanobacteria significantly increased the proportion of reproducing individuals in recipients, compared to a pre-exposure to green algae ($p < 0.0001$). *p* values were obtained from a Cox model for survival, and from a logistic regression for reproduction. Survival and reproduction data were obtained from the same individuals. Total sample size was $n = 360$ (4 donor genotypes × 2 donor diets × 3 recipient genotypes × 3 biological replicates × 5 individuals per replicate). On **d**, **e** and **f**, error bars correspond to 95% confidence limits

The monitoring of the gut microbiota composition in recipient *Daphnia* revealed a strong effect of time ($p = 0.001$, permutation MANOVA; Fig. 5a, b). Especially, Gammaproteobacteria dominated just after the transplant (day 1) and progressively decreased over time, while the proportion of Betaproteobacteria and Cytophaga-Flavobacteria progressively increased (Fig. 5a). The structure of the gut microbiota thus changes over the *Daphnia* life cycle. In addition, the relative contribution of the inoculum and of the recipient genotype to the gut microbiota structure varied over time ($p = 0.001$ and $0.029$, for the interaction time × inoculum and the interaction time × recipient genotype, respectively; permutation MANOVA; Fig. 5b). One day after the transplant (i.e., day 1), variation in gut microbial community composition were mainly explained by the inoculum effect, while the recipient genotype only had a marginally significant effect ($p = 0.001$ and $0.062$, respectively; permutation MANOVA; Fig. 5b). After 1 week (i.e., day 7), the structure of the gut microbiota had changed, and was mainly explained by the interaction between the inoculum and the recipient genotype ($p = 0.037$, permutation MANOVA; Fig. 5b), with the inoculum effect only being observed in *Recipient 1* ($p = 0.003$, permutation MANOVA). After 2 weeks (day 14), variation in the gut microbiota composition was explained by both the inoculum and the recipient genotype ($p = 0.001$ for both factors; Fig. 5b). Together, these results suggest that the bacterial community present in the habitat determines the establishment and the initial structure of the gut microbiota in *Daphnia*. Although this inoculum effect is still observed after 2 weeks, the structure of the gut microbiota is progressively reshaped, and starts to diverge between recipient genotypes, suggesting the occurrence of internal sorting processes.

Overall, the α-diversity also changed over time ($p < 0.0001$ and $p = 0.012$ for the operational taxonomic unit (OTU) richness and the Shannon diversity, respectively, ANOVA; Fig. 5c). The diversity was higher after 1 week (i.e., day 7) than at the other time points (OTU richness: $p < 0.0001$ for both day 7–day 1 and day 7–day 14 pairwise comparisons; Shannon diversity: $p = 0.015$ and $0.056$ for day 7–day 1 and day 7–day 14 pairwise comparisons, respectively; Tukey test; Fig. 5c). Intra-treatment variation (i.e., variation among replicates) in terms of both taxonomic composition and diversity tended to decrease over time, suggesting that the microbiota composition stabilized and converged (Fig. 5a, c).

## Discussion

Over the past few years, evidence is accumulating that the gut microbiota can be a crucial mediator of life history variation, as well as of acclimatization and adaptation to changing environmental conditions[1,8]. Predicting how, and to what extent, the gut microbiota may impact fitness requires to identify the links between variation in gut microbiota and host phenotype, and to understand how the microbiome communities are assembled[8]. Here we show that in the freshwater crustacean *Daphnia magna*,

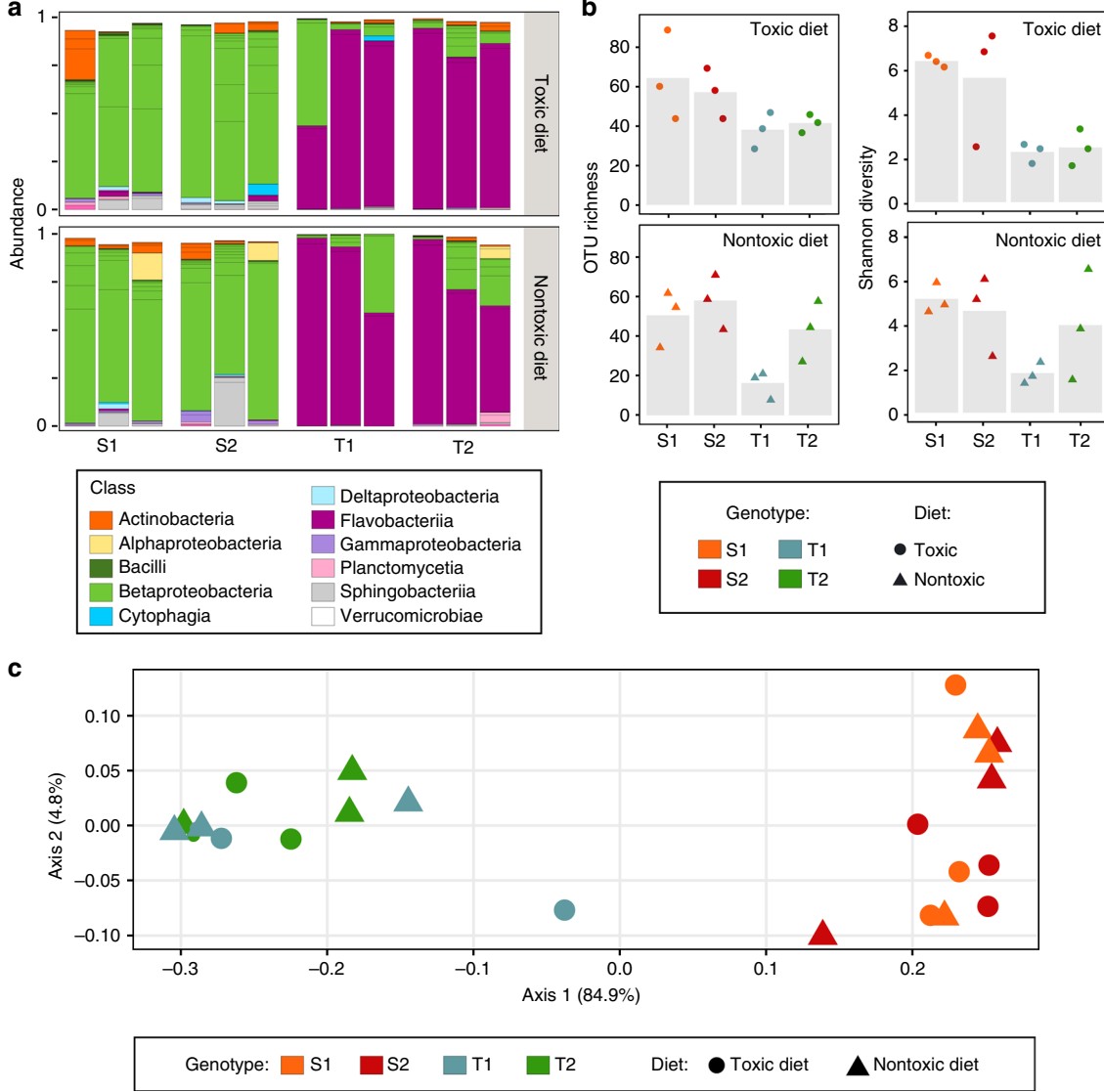

**Fig. 4** Effect of host genotype and diet on *Daphnia* gut microbiota structure. The gut microbiota of donor populations described in Figs. 2 and 3, corresponding to tolerant (*T1* and *T2*) and susceptible (*S1* and *S2*) genotypes, exposed to either a toxic cyanobacterial diet or a nontoxic green algal diet, was characterized through next-generation sequencing ($n = 3$ replicated populations per type of genotype × diet combination). For each population, the microbiota characterization was performed on a pool of 20 guts, obtained from adult individuals. **a** Relative abundance of OTUs in the gut microbiota of donor populations. Colors indicate the different bacterial classes. OTUs with an occurrence lower than 1% are not represented. **b** α-diversity (OTU richness and Shannon diversity) in the gut microbiota of donor populations. Bars indicate mean values, points indicate specific values for each population. Whereas α-diversity was not affected by the diet ($p = 0.16$ and $0.62$ for species richness and Shannon diversity, respectively), it significantly differed between tolerant and susceptible genotypes, with a higher diversity in susceptible genotypes ($p = 0.0006$ and $0.0002$ for species richness and Shannon index, respectively). The effect of genotype nested within tolerance class (i.e., tolerant or susceptible) was not significant ($p = 0.18$ and $0.32$ for species richness and Shannon index, respectively). *p* values were obtained from an ANOVA. **c** Principal component analysis of *Daphnia* gut microbiota in donor populations, using weighted UniFrac distance. Tolerant genotypes (*T1* and *T2*) on one hand, and susceptible genotypes (*S1* and S2) on the other hand, form two distinct clusters. Most of the variation in gut microbiota composition was explained by this tolerant vs. susceptible genotype effect ($p = 0.001$), while the diet did not have a significant impact ($p = 0.287$). The effect of genotype nested within tolerance class (i.e., tolerant or susceptible) was not significant ($p = 0.84$). *p* values were obtained from a permutation MANOVA

genotype-dependent gut microbiota drive tolerance to toxic cyanobacteria, which are responsible for harmful algal blooms in freshwater ecosystems. Our results further indicate that environmental conditions (i.e., exogenous microbial exposure and diet) and host genotype interact to shape not only the structure but also the functionality of the gut microbiota in this species.

Consistent with previous studies[21–23,25], we found interclonal variation in *Daphnia* tolerance to toxic cyanobacteria, indicating a high potential for a coevolutionary arms race between

cyanobacteria and their grazers. Susceptible genotypes showed a decrease in fitness when fed toxic cyanobacteria compared to nontoxic green algae, whereas tolerant genotypes maintained a high fitness in both diets. These results support the observations of Hairston et al.[22] and Jiang et al.[25] who showed genetic adaptation of *Daphnia* populations to cyanobacteria toxins in time and space, respectively. Interclonal variation in *Daphnia* tolerance to cyanobacteria, however, disappeared when *Daphnia* were made germ-free and received an identical microbial inoculum. Instead,

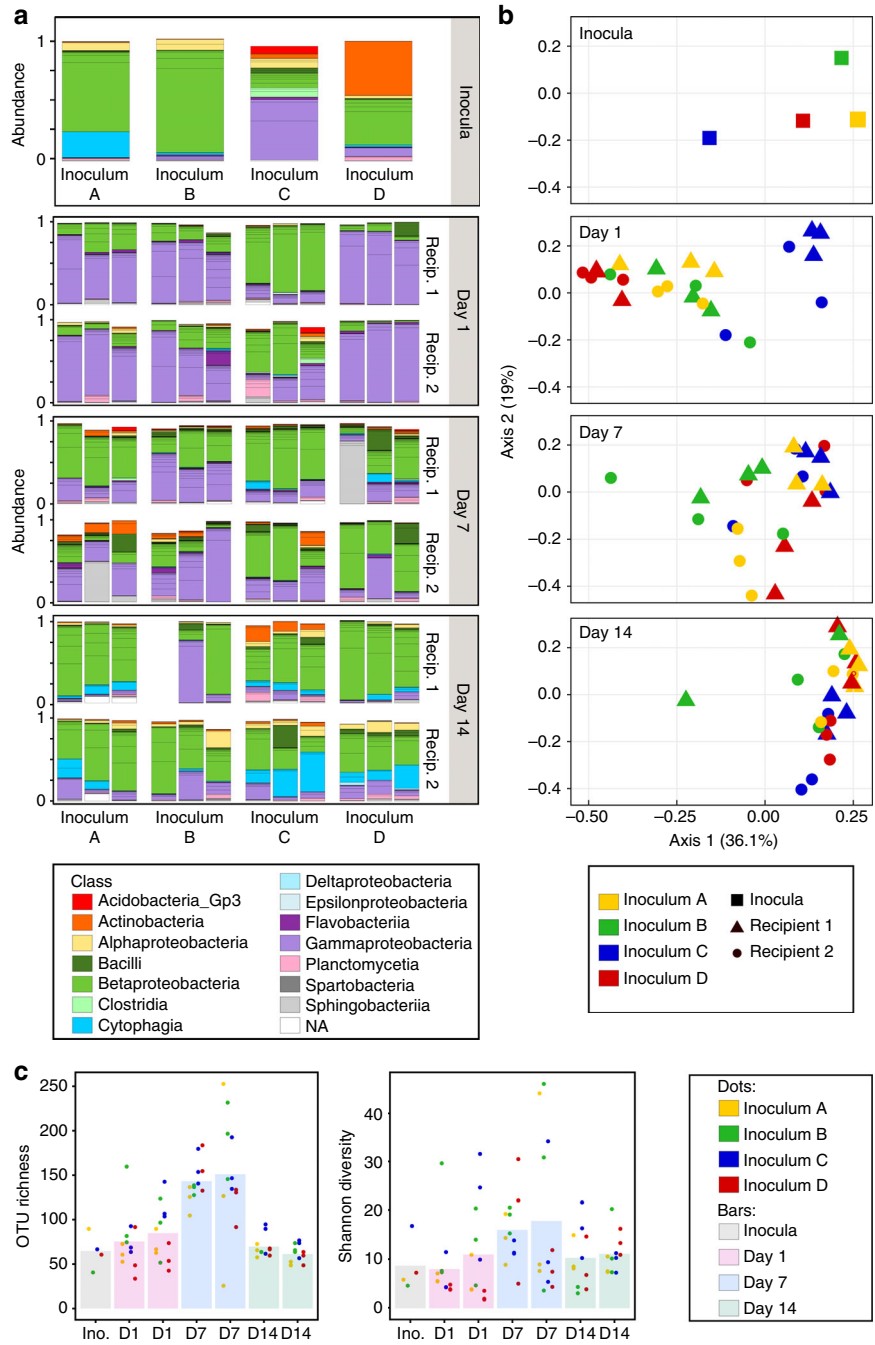

**Fig. 5** Host genotype and initial bacterial inoculum interact to shape *Daphnia* gut microbiota. **a** Relative abundance of OTUs in inocula (*n* = 4) and gut microbiota of recipient *Daphnia*, 1, 7, and 14 days after the transplant (*n* = 72 samples, i.e., 2 recipient genotypes × 4 inocula × 3 biological replicates × 3 time points; microbiota characterization was performed on pools of 10 guts). Colors indicate bacterial classes. OTUs with an occurrence lower than 1% are not represented. **b** Principal component analysis of microbial communities from inocula and *Daphnia* gut microbiota, using weighted Unifrac distance. The inocula and the different time points are represented on separate panels to facilitate interpretation. Gut microbiota composition in recipient *Daphnia* significantly changed over time (*p* = 0.001). At day 1, it was influenced by the inoculum but not by the recipient genotype (*p* = 0.001 and 0.062, respectively). At day 14, it was affected by both the inoculum and the recipient genotype (*p* = 0.001 for both factors). *p* values were obtained from permutation MANOVA. **c** α-diversity (OTU richness and Shannon diversity) in inocula and gut microbiota of recipient *Daphnia*. Bars indicate mean values, points indicate specific values for each sample. In gut microbiota, OTU richness and Shannon diversity changed over time (*p* < 0.0001 and *p* = 0.012, respectively). OTU richness and Shannon diversity were higher at day 7 than at other time points (day 1–7: *p* < 0.0001 and *p* = 0.015, respectively; day 7–14: *p* < 0.0001 and *p* = 0.056, respectively; day 1–7: *p* = 0.24 and 0.88, respectively). OTU richness and Shannon diversity were not affected by recipient genotype (*p* = 0.71 and 0.40, respectively). The inoculum had a significant impact on OTU richness (*p* = 0.015; pairwise comparisons: *p* = 0.077 and 0.092 for **A**–**C** and **C** and **D**, respectively; *p* > 0.9 for other pairwise comparisons), but did not impact Shannon diversity (*p* = 0.18). *p* values were obtained from an ANOVA, and Tukey tests for pairwise comparisons

variation among recipient *Daphnia* mirrored variation among their donors, with a higher tolerance in recipient *Daphnia* that received the gut microbiota from a tolerant donor genotype. Together, these results point to the gut microbiota as a major mediator of *Daphnia* tolerance to cyanobacteria, and suggest that differences in tolerance to cyanobacteria among *Daphnia* clones are mainly due to variation in gut microbiota composition. Supporting this hypothesis, our metagenetic analyses revealed that *Daphnia* genotype was an important determinant of gut microbiota composition and that tolerant and susceptible genotypes harbored very different gut microbial communities. We found that the gut microbiota of tolerant *Daphnia* genotypes was dominated by Flavobacteria. Flavobacteria are often associated with cyanobacterial blooms in freshwater ecosystems, with both their proportion[26,27] and their metabolic activity[28] usually increasing during *Microcystis* blooms. Members of the Cytophaga-Flavobacteria group cause the lysis of *Microcystis* cells and degrade dissolved organic matter derived from intracellular products of *Microcystis*[27,29]. The presence of Flavobacteria in the gut of *Daphnia* may therefore provide the host individuals access to otherwise inaccessible nutrients.

Our transplant experiment also revealed that the protective effect of the gut microbiota against *Microcystis* was enhanced after a prior exposure of the donors to cyanobacteria. This suggest that the microbiota shows plasticity upon exposure to different food qualities and that this may mediate *Daphnia*'s acclimatization to cyanoHABs[24]. The most common response of gut microbiota to changes in the host's diet or environmental conditions is a shift in the taxonomic composition of the community[2,3]. For example in humans, shifts from plant-based diet to meat-based diet have been followed by strong shifts in the gut microbial community, with an increase in animal protein-metabolizing bacteria and a decrease in bacteria that metabolize dietary plant saccharides[2]. Similarly, in a herbivorous rodent that can feed on highly toxic creosote bush, creosote toxins were shown to alter the population structure of the gut microbiota facilitating an increase in abundance of genes that metabolize toxic compounds[3]. Surprisingly, however, our metagenetic data did not reveal a significant effect of exposure to cyanobacteria on the taxonomic composition of the *Daphnia* gut microbiota. Given that we did observe an enhanced tolerance upon prior exposure of the donor strains to cyanobacteria, these combined data suggest that the *Daphnia* gut microbial community contains members that are physiologically versatile and could thus acclimate or genetically adapt to the cyanobacterial diet without changes in the taxonomic composition[15,30]. Bacteria often respond to environmental changes by expressing a range of metabolic capabilities, and therefore the existing community can confront new conditions through gene expression by individual cells[30]. In addition, as bacteria generally feature rapid growth and high mutation rates, and are capable of recombination via lateral gene transfer, members of the gut microbiota may also rapidly evolve and adapt to the diet, further enhancing the tolerance and the stability of the community[30].

The gut microbiota might affect *Daphnia* tolerance to cyanobacteria in different ways. First, gut microbes might contribute to the digestion of *Microcystis* cells. The importance of Flavobacteria in the microbiota of tolerant *Daphnia* supports this hypothesis in the present study. Such a role of symbiotic bacteria in host nutrition has been demonstrated in many species[1], including *Daphnia*[31]. A second potential route through which gut symbionts may help *Daphnia* to deal with cyanobacteria is the biodegradation of cyanobacterial toxins, which have negative impacts on *Daphnia* fitness[32]. Symbiont-mediated detoxification abilities have recently been shown to facilitate intake of plant toxins in herbivores[3] or pesticide resistance in insects[4], revealing the strong potential of the microbiota to increase its host's resistance to toxic environments. It is known that some free-living aquatic bacteria can degrade the microcystins released by cyanobacteria into nontoxic compounds[33]. A third way by which microbiota may mediate *Daphnia* resistance to cyanobacteria is through indirect epigenetic effects. The microbial metabolites produced by gut symbionts can modify epigenetic phenomena in host cells and thus alter the expression of host genes at the transcriptional level[34]. Recent studies have identified digestive enzymes[35] and transporter genes[32] that are assumed to regulate microcystin uptake in *Daphnia*. The expression of these genes was regulated as a specific response to microcystins, and the regulation was correlated with the level of tolerance of the tested clones[32]. The microbiota might affect the expression of these genes, thus acting indirectly on *Daphnia* responses to cyanobacterial toxins. This hypothesis, however, remains to be tested.

In the present study, we mainly observe differences in microbiota community composition between susceptible and tolerant *Daphnia* clones, and no changes in community composition upon exposure to cyanobacteria. We have indications, however, that there may be variation on this theme, as in a pilot experiment we did obtain evidence for a difference in microbiota community composition between populations exposed to edible green alga and toxic cyanobacteria (Supplementary Fig. 4). In this pilot experiment the metagenetic analysis was limited (i.e., a single replicate per genotype × diet combination), but suggested a strong impact of diet. Especially, when *Daphnia* were exposed to toxic cyanobacteria, the gut microbiota of both tolerant and susceptible genotypes tended to be more diverse, and enriched in bacterial groups (e.g., *Sphingomonas* sp, *Pseudomonas* sp., and *Phenylbacterium* sp.) known to degrade microcystin (i.e., the main toxin produced by the cyanobacteria *M. aeruginosa*), or to possess a gene involved in the microcystin degradation pathway[33,36]. This suggests that gut bacteria may contribute to *Daphnia* tolerance to cyanobacteria through their detoxification capabilities. Flavobacteriaceae, which were found to dominate the gut of tolerant genotypes in the present, fully replicated study, were not detected in the pilot test. We can only speculate on what caused the differences in microbial community composition between the two experiments, but a possible explanation is that the environmental pools of microbes to which the *Daphnia* were exposed were different. Prior to both experiments, all *Daphnia* clones were regularly exposed to the same batches of pond water in order to provide them with a diverse pool of potential symbionts. The characteristics of the pond water may strongly vary over time (e.g., occurrence of algal blooms), hence the initial microbial inoculum likely differed strongly among experiments, impacting the structure and the dynamics of the gut microbiota. In the present study, the pond water was added less often compared to the preliminary test, hence the microbial pool of microbes available to the *Daphnia* might have been less diverse, and this might have impeded a response of the gut microbiota to a shift in diet at the taxonomic level[15,30]. Alternatively, such a response was not necessary given that the communities were already dominated by Flavobacteria. The overall perspective offered by the combined data suggests that tolerance to cyanobacteria in *Daphnia* does not rely on specific host-associated symbionts, and that *Daphnia* can build different tolerant microbial communities, depending on the pool of bacteria available in the surrounding environment.

The experiment in which we monitored the gut microbiota dynamics following a transplant revealed that the taxonomic composition was dependent on both host genotype and environmental factors, and varied across the *Daphnia* life cycle. In juveniles, the gut microbiota structure was mainly determined by exogenous microbial exposure, i.e., the pool of symbionts

available in the environment. The taxonomic composition of the microbial community present in the *Daphnia* gut was nevertheless very different from that of the inoculum, indicating a selective recruitment of symbionts during the colonization process. A selective recruitment of bacteria was previously observed in mice, in which bacteria from diverse habitats were shown to colonize the gut, but despite these highly dissimilar input communities, the established output communities tended to converge[10]. Our result is also consistent with a previous study in *Daphnia* showing that the microbial community established in the gut is different from that of the cultivation water[37]. In our analysis, the taxonomic composition of the microbiota progressively changed over a time period of 2 weeks, as the host became adult, with a decrease in the proportion of Gammaproteobacteria, and an increase in the proportion of Betaproteobacteria and Cytophaga-Flavobacteria. The α-diversity also changed over time, with a peak 7 days after exposure to the inoculum. Such temporal variation in gut microbiota structure has been observed in other species, from insects[38] to humans[39], and may act as a buffer against variation in metabolic demands across the life cycle[1]. Alternatively, these changes may be due to the colonization process, which can be viewed as an ecological succession, in which the initial adhesion of early colonizers shapes the metabolic milieu in a manner permissive for establishing a more diversified collection of bacterial species, and in which syntropic and competitive interactions between community members may generate strong priority effects[1,10,40]. The gut microbiota of *Daphnia* is thus expected to experience high spatiotemporal variation, also depending on the composition of the bacterioplankton in the surrounding environment (i.e., microbial availability), which may explain the inconstancy in gut microbiota composition among studies in the literature[16,37,41].

While the gut microbiota composition in juveniles was mainly determined by microbial exposure, it evolved differently between genotypes in later life stages. These results are consistent with the genetic variation in gut microbiota composition observed in other species, including mice[42] and humans[13]. The strength of our approach is that the clonal reproduction of *Daphnia* allowed us to quantify the relative contribution of host genotype and inoculum to gut microbiota composition. Hence, even if the gut microbiota is acquired independently each generation from environmental bacteria or horizontal transfer, the host genetic background has an effect on the recruitment of these bacteria, and on their persistence in the digestive tract. Selection on symbiont-mediated traits promoting host performance under local environmental conditions (e.g., exposure to cyanobacteria) may result in an evolution of host genes involved in acquisition, control, and tolerance of beneficial symbionts, allowing for an indirect co-inheritance of nuclear genes and microbes[1,14]. Such genetic control over the gut microbiota likely occurs through immunity. Immunity is the guardian of the host gut environment. It coordinates cellular and biochemical responses through the epithelial cells, ensuring the elimination of pathogens while maintaining a coexistence with mutualistic symbionts[1,5,43]. Recent studies show that the host's immune response, especially the production of antimicrobial peptides, play an important role in shaping the gut microbiota[44]. The host's immune response was shown to vary genetically[45], including in *Daphnia*[46]. Hence, *Daphnia* may assemble different microbial communities, depending on their immune profile. Such immune variation may explain the difference in gut microbiota composition between tolerant and susceptible genotypes. The lower diversity observed in tolerant genotypes may result from a higher selectivity in the recruitment of bacteria in the gut.

Our study opens a plethora of questions for future research. First, it will be important to quantify the contribution of host genotype-mediated microbiota to the spatial[25] and temporal[21] adaptation patterns observed in natural *Daphnia* populations subjected to varying levels of cyanobacterial stress. Second, future studies should investigate the mechanisms by which the microbiota confers tolerance to cyanobacteria. More specifically, our study raises the question whether the microbiota of tolerant host populations can metabolize the cyanotoxins. Our sequencing results of the pilot experiment hinted at this, but we observed no evidence for this in terms of the presence of specific taxa known to degrade cyanotoxins in our main experiment. Sadler and von Elert[47] did not find evidence for a biodegradation of microcystins during the digestion of *Microcystis* cells in *Daphnia*. The occurrence of microcystin degradation may, however, depend on a combination of factors, including e.g., the bacterial community available to the *Daphnia*, the *Daphnia* genotype, or the duration of *Daphnia* exposure to the cyanotoxins. Assays that directly quantify the capacity of gut microbiota to degrade cyanotoxins are an important avenue for further work, as will be mono-association tests to assess the precise functions of specific taxa[48]. Third, a key observation of our study is that host genotype mediates microbiota community structure in *Daphnia*. Obtaining insight in the role of *Daphnia* immunity in structuring the gut microbiota, and comparing the immune profile of *Daphnia* among genotypes (e.g., tolerant vs. susceptible) and diets (e.g., toxic vs. nontoxic), will be key[44]. This would enable us to better understand not only the host immune modulation that contributes to shape the tolerant phenotype of gut microbiota against cyanobacteria but also the strategy in *Daphnia*'s innate immunity (immune tolerance or resistance) against toxic cyanobacteria themselves. At last, to understand the evolutionary dynamics of host–microbiota associations in *Daphnia*, it would be important to determine how far the symbionts draw benefits from this relationship. Collecting and freezing microbiota at different points during a host–symbiont coevolutionary experiment, together with reciprocal microbiota inoculations, could help assess local adaptation of symbionts to their host following the general methodology of Red Queen studies[1].

According to the hologenome concept[49], adaptation of organisms to novel environments does not only result from the interaction of the host genome with the environment, but rather from the interaction of their hologenome (i.e., the cumulative genomes of the host and its associated symbionts) with the environment. Although this hypothesis is gaining much interest[14], strong empirical support is still lacking. Our study shows that (1) the gut microbiota is dependent on both host genotype and environmental factors, and (2) both host-genotype mediated and environmentally mediated variation in gut microbiota structure affect host fitness and its response to environmental pressure, here exposure to toxic cyanobacteria. Both observations represent essential prerequisites for selection to act on the hologenome[1,14]. In the more specific context of responses of aquatic systems to cyanoHABs, a rapidly increasing component of global change[50], we show that in the pivotal grazer *Daphnia*, tolerance to toxic cyanobacteria is mediated by the gut microbiota, whose structure is partially mediated by host genotype. In a seminal resurrection ecology study, Hairston et al.[21] found strong inter-clonal variation in *Daphnia* tolerance to cyanobacteria, and convincingly demonstrated that tolerance of *Daphnia* genotypes to cyanobacteria increased over time in response to increasing eutrophication. In this study, *Daphnia* genotypes were hatched from resting eggs, and exposed to identical microbial pool throughout the experiment, providing a strong argument in favor of the importance of host genotype. In light of our results, one might speculate that these genotypes had acquired different gut microbiota, e.g., through selective recruitment mechanisms, which then lead to differences in tolerance. This interaction

between host genotype, microbiota, and performance under cyanobacteria blooms suggests that genotype-mediated symbiont community structure is an important mediator of the genetic mosaic of coevolution between toxic cyanobacteria and their grazers[23], and a key determinant of how freshwater ecosystems respond to climate warming. Our study shows that the gut microbiota in *Daphnia* acts as an extended phenotype of the *Daphnia* genotype that increases the capacity of *Daphnia* hosts to cope with cyanobacteria, and might represent the key phenotype that modulates adaptation to cyanoHABs in this zooplankton species.

## Methods

**D. magna genotypes**. Six clones of *D. magna* were used in our experiments: OM2NF2 (named "*S1*" in the text, to easily identify that it is a genotype susceptible to *Microcystis*); B7 ("*S2*"); KNO15.04 ("*T1*"); T9 ("*T2*"); BSW7 ("*Recipient 1*"); and OM2F8 ("*Recipient 2*").

Genotypes B7 ("*S2*") and T9 ("*T2*") were originally isolated from an 8.7 ha shallow, manmade, pond located in Oud Heverlee, in Belgium. This pond was constructed in 1970 for fish culture[51]. Clonal lineages were established from resting eggs sampled in two sediment core sections: bottom (18–21 cm depth range, corresponding to the 1970–1972 period) and top (3 cm, corresponding to ca 1988) for B7 and T9, respectively. The genotype KNO15.04 ("*T1*") was isolated from a small pond (350 m²) located near Knokke, at the Belgian coast (51°20′05.62″ N, 03° 20′53.63″ E), and characterized as a fishless, mesotrophic lake (concentration suspended matter: 2.52 mg/L and chlorophyll a: 6.67 µg/L). Genotypes OM2NF2 ("*S1*") and OM2F8 ("*Recipient 2*") were isolated from a 3.7 ha inland pond located in Heverlee (Oude Meren, Abdij van' t Park; 50°51′47.82″ N, 04°43′05.16″ E), in Belgium. This pond contains fish and is considered as eutrophic (concentration suspended matter: 43.50 mg/L and chlorophyll a: 215.45 µg/L). KNO15.04, OM2NF2, and OM2F8 clonal lineages were established from resting eggs collected from the upper two centimeters of the lake sediment. Genotype BSW7 ("*Recipient 1*") was isolated from Bysjön lake in Sweden. The history of *Microcystis* occurrences of the different ponds is not documented, except for the Heverlee-Oude Meren pond (OM2NF2 and OM2F8) and the Bysjön lake (BSW7), in which *Microcystis* blooms have been observed (L.D.M., K.U. Leuven, personal observation).

All genotypes were maintained in the laboratory under standardized conditions for several years prior to the experiment. Stock *Daphnia* clonal lineages were cultured in re-constituted freshwater (ADaM medium)[52], at a temperature of 19 ± 1 °C and under a 16:8 h light:dark cycle, in 500 mL glass jars (at a density of 20 individuals/L). They were fed daily with saturating amounts of the green algae *S. obliquus*. Medium was refreshed once a week.

As all genotypes were hatched in the laboratory from resting eggs, and maintained in the laboratory for several years, it is unlikely that the experimental populations contain bacteria from the pond of origin.

**Cultivation of green algae and cyanobacteria**. The unicellular green algae *S. obliquus* (strain CCAP 276/3A, provided by the Culture Collection of Algae and Protozoa, UK) and the unicellular cyanobacteria *M. aeruginosa* (strain PCC 7806, toxic strain producing microcystin LR, provided by the Pasteur Culture Collection, Institut Pasteur, Paris, France) were used to feed the *Daphnia*. The freshwater green alga *S. obliquus*, which has been shown to have a high nutritional value for cladoceran zooplankton, is commonly considered as a standard good-quality food for *Daphnia*, and is therefore very commonly used in culture experiments[53]. In contrast, the cyanobacteria *M. aeruginosa* is a (relatively) poor food for cladocerans because of their low levels of highly unsaturated fatty acids and sterol contents, as well as their low digestibility[54,55]. In addition, many *M. aeruginosa* strains, including the one used in the present study, produce toxins and bioactive compounds, such as microcystins and protease inhibitors[18,56], which are detrimental for *Daphnia*[57]. *Microcystis* ssp. are the most commonly reported species responsible for toxic cyanobacterial blooms in lakes and ponds across the world[18].

*S. obliquus* and *M. aeruginosa* were grown in WC medium[58] and modified WC medium (WC medium without Tris), respectively. They were cultured under sterile conditions in a climate chamber at 20 ± 2 °C with a light:dark cycle of 16:8 h, in 2 L glass bottles, with constant stirring and aeration. Filters (22 µm) were placed at the input and the output of the aeration system to avoid any bacterial contamination. The algae were weekly harvested in stationary phase. Axenity of the cultures was checked on LB medium agar plates. Ash-free dry weight of the cultures was determined following Moheimani et al.[59].

**Effects of gut microbiota transplants on cyanobacterial tolerance**. This is experiment 1.

To determine the role of gut microbiota in *Daphnia* tolerance to toxic cyanobacteria, we performed reciprocal gut microbiota transplants between tolerant and susceptible genotypes, and subsequently monitored the survival and reproduction of recipient *Daphnia* upon cyanobacterial exposure. The donor genotypes were prior to the transplant exposed to either a cyanobacterial or a green

algal diet for several months, which further allowed us to assess the effects of a cyanobacterial pre-exposure of the donors on the gut microbiota functionality.

*Evaluation of cyanobacterial tolerance in experimental genotypes*: For this experiment, we chose four genotypes (T9, KNO15.04, OM2NF2, and B7) that were in pilot experiments observed to respond differently to cyanobacterial exposure in the laboratory. Their tolerance level was assessed by exposing them to either a toxic cyanobacterial or a nontoxic green algal diet, and determining their survival rate.

Three iso-female lines of each genotype were grown separately in 2 L jars, under the same conditions as described above, for two generations, to control for maternal effects. Then, for each maternal line, adult individuals of the same generation (n = 10), which produced their second brood, were placed in fresh ADaM medium to release their juveniles. Fourteen 1-day-old juveniles were subsequently randomly sampled from each maternal line, and individually placed in 50 mL conical centrifuge tubes containing 45 mL of ADaM medium. Half of the *Daphnia* were attributed to a "nontoxic green algae" treatment, which consisted in a 100% *S. obliquus* diet, while the other half was attributed to a "toxic cyanobacteria" treatment, which consisted of a 80% *M. aeruginosa*–20% *S. obliquus* diet. Hence, for each genotype, a total of 21 individuals were tested in each treatment. Algae were provided every 2 days, with a final carbon concentration of 1 mg C/L. Survival was monitored every 2 days, during 2 weeks.

Two genotypes (KNO15.04 and T9) were found to have similar survival rates in both treatments, and were qualified as tolerant genotypes (hereafter designed as T1 and T2, respectively). The other two genotypes (OM2NF2 and B7) were found to have lower survival rates under the toxic cyanobacteria treatment and were qualified as susceptible genotypes (hereafter designed as S1 and S2, respectively).

*Exposure of donors to green algal vs. cyanobacterial diet*: For each genotype, three iso-female lines were grown separately in 2 L jars, under similar conditions as described above, on a diet of saturating amounts of *S. obliquus*. When a sufficient number of individuals was reached, 120 juveniles were sampled in each iso-female line, and divided into two 2 L jars (each containing 60 individuals). The first jar was fed a green algal diet, while the second one was fed a cyanobacterial diet (Fig. 2). This way, we obtained a total of 24 populations (4 genotypes × 2 diets × 3 replicates). Algae were provided every 2 days, with a final carbon concentration of ~1.5 mg C/L. The green algal diet was composed of 100% *S. obliquus*, while the cyanobacterial diet was composed of a mixture of *M. aeruginosa* and *S. obliquus*. The ratio *Microcystis/Scenedesmus* was not constant over time, and was adjusted depending on the condition of the *Daphnia*: if *Daphnia* suffered too much from the presence of *Microcystis*, resulting in a too high mortality, the ratio was reduced, but always ranged between 50% *Microcystis*–50% *Scenedesmus* and 80% *Microcystis*–20% *Scenedesmus*. The ratio was kept the same among all jars. Medium was refreshed once a week. Water from a pond situated in the ECOLAB on the campus (Kortrijk, Belgium, 50°48′30.3″ N, 3°17′38.0″ E) was added to the ADaM medium (15% of the final volume) every 2 weeks, in order to provide a large diversity of bacteria and optimal growth conditions for the *Daphnia*.

*Extraction of the donors' microbiota*: After 6 months (i.e., 12 generations) of exposure to the two types of diet, 15 *Daphnia* were sampled from each of the 24 donor populations and placed in sterile (i.e., autoclaved) ADaM medium during 24 h, to remove food particles from the gut. Their guts were subsequently extracted with dissection needles under a stereomicroscope, placed together in 1.5 mL Eppendorf tubes containing 1 mL of deionized sterile water, and crushed with a pestle. This way, we obtained a total of 24 inocula (4 genotypes × 2 diets × 3 replicates).

*Preparation of recipient Daphnia and gut microbiota transplant*: Recipient *Daphnia* were obtained from T9, KNO15.04, and OM2NF2 genotypes. The genotype B7 was discarded because the number of individuals available at the moment of the experiment was too low. For each genotype, five iso-female lines were grown separately in 2 L jars for two generations, under similar conditions as described above, on a diet of saturating amounts of *S. obliquus*. Females carrying parthenogenetic eggs (second brood) were dissected under a stereomicroscope and their eggs were collected in a Petri dish containing ADaM (n = 50 eggs per genotype and per iso-female line). Only recently deposited eggs, which are characterized by the presence of an external membrane, were isolated. These eggs were then disinfected in order to obtain germ-free *Daphnia*, using the protocol developed by Callens et al.[16]. This manipulation was performed under sterile conditions, in a laminar flow hood. To remove microbial organisms from the eggs, they were placed in a Petri dish containing 10 mL of a 0.01% peracetic acid (PAA) solution. The Petri dishes were gently agitated for 10 min, after which the eggs were transferred to another Petri dish containing sterile ADaM to remove any PAA residues. This rinsing step was repeated, to ensure that any trace of PAA was removed. Afterwards, eggs were transferred into six-well (cell culture) sterile plates, each well containing 8 mL of sterile ADaM and about 40 eggs, and incubated at 20 ± 0.5 °C. Eggs were allowed to hatch during 48 h under sterile conditions, and the resulting germ-free juveniles were used as recipients in the transplant experiment.

Twenty-four germ-free *Daphnia* juveniles were then sampled from each iso-female line, and individually placed in 50 mL centrifuge tubes containing 45 mL of sterile ADaM. Each of these 24 *Daphnia* received one of the 24 microbiota inocula by adding a microbiota extract (i.e., the equivalent of one crushed gut) to each tube, as well as 0.5 mg C/L of the green alga *S. obliquus*. *Daphnia* were left in these conditions during 2 days, to ensure that they ingested enough bacteria from the microbiota extracts, and were subsequently transferred in fresh sterile ADaM medium. Then, they were subjected to a cyanobacterial diet (80% *Microcystis*–20%

*Scenedesmus*), with a final carbon concentration of 1 mg C/L. Algae were provided every day, and the medium was refreshed twice a week. Survival and reproduction were monitored every day, during 2 weeks. As in some treatments, a large proportion of individuals died before reproducing, we did not analyze the fecundity data, but only the proportion of individuals that reproduced. These tests were performed on a total of 360 recipient *Daphnia* (4 donor genotypes × 2 donor diets × 3 replicated donor populations × 3 recipient genotypes × 5 individuals).

Statistical analyses: Analyses of cyanobacterial tolerance data were performed using the SAS 9.4 software. Survival was analyzed with a Cox proportional hazards model regression (PHREG procedure). For the evaluation of cyanobacterial tolerance in the experimental *Daphnia* genotypes, genotype, diet, and their interaction were chosen as fixed factors, while the random effect of maternal line (nested within genotype) was taken into account through the ID statement, which allows to adjust for intracluster correlation. For the effects of gut microbiota transplants on tolerance to cyanobacteria, Recipient genotype, donor genotype, and donor diet, as well as their interactions, were chosen as fixed factors, while maternal line (nested within recipient genotype) and replicated population (nested within donor genotype and donor diet) were taken into account through the ID statement. In both analyses, nonsignificant interactions were removed from the model when their *p* value was above 0.05. The survival times of individuals that were still alive at the end of the experiment were coded as censored. As ties in survival times were numerous, we used the approximate likelihood of Efron. To check for the proportionality assumption, constructed time-dependent variables (i.e., interaction terms that involve time and predictor variables) were included in the model, and kept when they were significant[60]. Pairwise comparisons were performed using the CONTRAST statement, which provided both the hazard ratios between groups for the variable of interest, and the associated *p* values. The survival curves were obtained using the LIFETEST procedure.

In the transplant experiment, the proportion of recipient *Daphnia* that reproduced before dying was analyzed with a logistic regression (LOGISTIC procedure). In some treatments, because reproducing *Daphnia* were very rare, all the observations had the same event status, resulting in separation phenomenon. Hence, a classic logistic regression model could not be applied. To overcome this problem of separation, we used the Firth's penalized likelihood approach[61]. The response variable was the proportion of reproducing individuals (per recipient genotype, donor genotype, donor diet, and population (nested within donor genotype and donor diet)). Recipient genotype, donor genotype, donor diet, as well as their interactions, were chosen as fixed factors. Pairwise comparisons were performed using the CONTRAST statement.

**Gut microbiota composition in tolerant and susceptible genotypes**. This is experiment 2.

To determine whether the composition of the gut microbiota differed between tolerant and susceptible genotypes, and whether it was affected by exposure to a cyanobacteria, the gut microbiota of the 24 donor populations described above was characterized through next-generation sequencing of 16S rRNA. These 24 donor populations were the same as those described in experiment 1, which were continuously maintained in the laboratory, under similar conditions as in experiment 1. The gut microbiota characterization of these populations was performed ~1.5 year after the transplant, i.e., after ~58 generations of exposure to cyanobacterial vs. green algal diet.

Twenty adult *Daphnia* were sampled from each donor population and placed in fresh, sterile (i.e., autoclaved) ADaM medium during 24 h, to remove most transient bacteria and food particles from the gut. Their guts were dissected and pooled, and the microbiota was extracted, using the same protocol as described in experiment 1. The 24 samples were stored at −20 °C until further processing.

Sequencing library preparation: DNA was extracted using a PowerSoil DNA isolation kit (MO BIO laboratories), and dissolved in 20 μL MilliQ water. The total DNA yield was determined using a Qubit dsDNA HS assay (Invitrogen) on 3 μL of sample. Because of initially low bacterial DNA concentrations in some samples, a nested PCR was applied to increase specificity and amplicon yield[11,62]. The full-length 16S rRNA gene was first amplified with primers 27F and 1492R on 10 ng of template (94 °C—30 s; 50 °C—45 s; and 68 °C—90 s; 30 cycles) using a high-fidelity Pfx polymerase (Life Technologies). PCR products were subsequently purified using the QIAquick PCR purification kit (Qiagen). To obtain dual-index amplicons of the V4 region, a second amplification was performed on 5 μL of PCR product using primers 515F[63] and a slightly modified version of primer 806R to increase detection of SAR11 bacterioplankton[64] for 30 cycles (94 °C—30 s; 55 °C—30 s; and 68 °C—60 s). Both primers contained an Illumina adapter and an 8-nucleotide (nt) barcode at the 5′-end. For each sample, PCRs were performed in triplicate, pooled, and gel-purified using the QIAquick gel extraction kit (Qiagen). An equimolar library was prepared by normalizing amplicon concentrations with a SequalPrep Normalization Plate (Applied Biosystems) and subsequent pooling. Amplicons were sequenced using a v2 PE500 kit with custom primers[63] on the Illumina Miseq platform (KU Leuven Genomics Core), producing 2 × 250-nt paired-end reads. In this way, we generated 24 samples representing 4 genotypes × 2 diets × 3 replicates.

Processing of sequencing data: Sequence reads were processed using R 3.3.2 (R Core Team, 2016) following Callahan et al.[65]. Sequences were trimmed (the first 10 nucleotides and from position 180 onwards) and filtered (maximum of 2 expected errors per read) on paired ends jointly. Sequence variants were inferred using the high-resolution DADA2 method, which relies on a parameterized model of

substitution errors to distinguish sequencing errors from real biological variation[66]. Chimeras were subsequently removed from the data set. After filtering, the average number of reads per sample was 37,549 (minimum = 14,557 reads and maximum = 66,460 reads). Taxonomy was assigned with a naive Bayesian classifier using the RDP v14 training set. OTUs with no taxonomic assignment at phylum level or which were assigned as "chloroplast" or "cyanobacteria" were subsequently removed from the data set. The final data set contained a total of 894,270 reads, with on average 37,261 reads per sample (minimum = 14,504 reads and maximum = 66,453 reads).

Analysis of sequencing data: As measures for α-diversity within the different microbial communities, OTU richness (total number of OTUs present) and Shannon index (taking into account both OTU richness and the relative abundance of OTUs) were calculated using the vegan package in R. The effects of diet (i.e., toxic cyanobacteria vs. nontoxic green algae), tolerance profile (i.e., tolerant vs. susceptible), and genotype (nested within tolerance profile) on OTU richness and Shannon index were assessed through ANOVA. Interactions between the different factors were included in the initial model, and removed when nonsignificant (*p* > 0.05).

To investigate differences in community composition (β-diversity) between the different microbial communities, weighted UniFrac distances were calculated[67] and plotted using principal coordinates analysis with the phyloseq package in R[68]. The effects of diet, tolerance profile, genotype (nested within tolerance profile), and their interactions on β-diversity was assessed through a permutation MANOVA, using the Adonis function of the vegan package in R.

To identify the bacterial families that differed between tolerant and susceptible genotypes, OTUs were grouped at the family level, and families representing <1% of the reads were discarded. Differential abundance analyses were then performed with the Bioconductor package DESeq2[69].

**Assembly and temporal dynamics of the *Daphnia* gut microbiota**. This is experiment 3.

To investigate more deeply the dynamics of the gut microbiota composition after a transplant, and to determine the relative contribution of colonization (i.e., exogenous microbial exposure) vs. internal, genotype-dependent sorting processes to the gut microbiota structure, we exposed germ-free *Daphnia* of two recipient genotypes to four types of microbial inocula, and monitored the gut microbiota composition over time by next-generation sequencing on 16S rRNA. The clones BSW7 (Recipient 1) and OM2F8 (i.e., Recipient 2) were randomly chosen as "recipient" genotypes, among *Daphnia* clones available in the laboratory at that moment (some of the clones used in experiment 1 and 2 had been lost prior to the starting of experiment 3). The four types of microbial inocula were obtained by crushing the guts of *Daphnia* from four distinct populations, corresponding to two clones (T9 and BSW7), each one exposed to two combinations of food and bacterioplankton. After the transplant, the gut microbiota composition in recipients was monitored over time (at three different time points corresponding to 1, 7 and 14 days) using next-generation sequencing on 16S rRNA.

Preparation of the microbial inocula: The *Daphnia* clones T9 and BSW7 were grown in 2 L jars (containing ~40–60 individuals), with 6 maternal lines (i.e., 6 jars) per clone. For each clone, the 6 maternal lines were divided into 2 groups, which were submitted to different experimental conditions in order to maximize variation in their gut microbial communities and obtain different types of microbial inocula for the transplant experiment. In the first condition, *Daphnia* were fed the green algae *S. obliquus*, and filtered water from a pond situated on the campus (Kortrijk, Belgium, 50°48′30.3″ N, 3°17′38.0″ E) was added to the ADaM medium (5% of the final volume) approximately every 3 weeks to provide *Daphnia* with bacterioplankton. In the second condition, *Daphnia* were fed the toxic cyanobacteria *M. aeruginosa*, and water from another pond of the campus was added to provide *Daphnia* with another set of environmental bacteria. Algae were provided every 2 days, with a final carbon concentration of circa 1.5 mg C/L. In the second condition, the green algae *S. obliquus* was added regularly, to avoid any crash of our populations. This way, we obtained 4 types of population (2 genotypes × 2 conditions) that were expected to harbor different gut microbial communities, with three replicates per type of population.

After several weeks, *Daphnia* from these populations were isolated in order to extract their gut microbial community. For each type of population, 60 adult *Daphnia* (i.e., 20 *Daphnia* per maternal line) were sampled, and placed together in fresh, sterile (i.e., autoclaved) ADaM medium during 24 h to remove food particles from the gut. Their guts microbiota was then extracted using the same protocol as in experiment 1. This way, we obtained 4 microbial inocula, each one containing the gut microbiota of 60 *Daphnia*. The inocula "A", "B", "C" and "D" correspond to the gut microbiota of "BSW7 genotype × condition 2", "BSW7 genotype × condition 1", "T9 genotype × condition 2" and "T9 genotype × condition 1", respectively.

Preparation of recipient *Daphnia* and microbiota transplant: Twelve iso-female lines were created for the clones OM2F8 and BSW7 and grown separately in 500 mL jars for several generations, under similar conditions as described above, on a diet of saturating amounts of *S. obliquus*. Females carrying parthenogenetic eggs were dissected and their eggs (*n* = 40 eggs per genotype and per iso-female line) were collected and made germ-free, as described in experiment 1. Eggs were allowed to hatch during 48 h, and the resulting germ-free juveniles were used as recipients in the transplant experiment.

For each recipient clone, 40 germ-free juveniles were sampled from each maternal line, and placed together in a 50 mL centrifuge tube containing 45 mL of sterile ADaM. The 12 tubes were then divided into 4 groups, each one inoculated with 166 µL of either "A", "B", "C", or "D" microbial inoculum (i.e., the equivalent of 10 crushed guts, resulting in an average of 1 crushed gut for 4 germ-free recipient *Daphnia*), as well as 1 mg C/L of the green alga *S. obliquus*. *Daphnia* were left in these conditions during 2 days, to ensure that they ingested enough bacteria from the microbiota extracts, and were subsequently transferred in a bigger jar (500 mL) containing fresh sterile ADaM medium. They were submitted to a cyanobacterial diet (80% *Microcystis*–20% *Scenedesmus*), with a final carbon concentration of 1 mg C/L. Algae were provided every day. There were thus a total of 24 recipient populations (2 recipient genotypes × 3 replicated iso-female lines × 4 inocula), with 40 *Daphnia* juveniles in each population.

Characterization of gut microbiota and sequencing data analysis: Ten *Daphnia* were sampled from each recipient populations at three different time points ($t = 0$, 7 days, and 14 days) in order to monitor the gut microbiota composition over time. These *Daphnia* were first placed in jars filled with 200 mL of sterile (i.e., 0.22 µm filtered) ADaM medium during 24 h, to remove most non-symbiotic transient bacteria and food particles from their gut. *Daphnia* guts were subsequently dissected under a stereomicroscope using sterilized dissecting needles, and pooled in a microcentrifuge tube with 10 µL of sterile MilliQ water. The 72 resulting samples (2 recipient genotypes × 4 inocula × 3 replicates × 3 time points) were immediately frozen and stored at −20 °C until processing.

The sequencing library preparation and the processing of sequencing data were performed as described in experiment 1. The average number of reads per sample (after filtering) was 51,176 (minimum = 11,979 reads and maximum = 150,411 reads).

α-diversity (OTU richness and Shannon index) and community composition divergence (β-diversity; weighted UniFrac distances) were calculated and analyzed as described in experiment 2. The effects of time (Day 1, 7 and 14), inoculum (A, B, C, or D), and recipient genotype (*Recipient 1* and *Recipient 2*) on α- and β-diversity were tested through an ANOVA and a permutation MANOVA, respectively. Interactions between the different factors were included in the initial models, and removed when nonsignificant ($p > 0.05$). For alpha-diversity, a Tukey honest significant difference test was performed for post hoc comparison.

**Data availability**. Sequence data have been deposited in the Sequence Read Database (SRA) under project IDs SRP115642 (BioProject PRJNA398629) and SRP115678 (BioProject PRJNA398630) for experiment 2 and experiment 3, respectively. All other relevant data are available in this article and its Supplementary Information files, or from the corresponding authors upon request.

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

## Acknowledgements

Funding was provided by the research projects: FWO G.0643.13, KU Leuven Centre of Excellence SEEDS PF/2010/07, IAP Belspo project SPEEDY, and Marie Curie Intra-European Fellowship No. FP7-PEOPLE-2012-IEF-329870-ANSWER (to E.M., currently FWO postdoctoral fellow). We are grateful to Isabel Vanoverberghe for her assistance during experiments, and to Gregory Maes and Jeroen Van Houdt of the KU Leuven genome center for their feedback on the sequencing. We thank Koenraad Muylaert, Jeroen Raes, Shira Houwenhuyse, and Dieter Ebert for stimulating discussions. We thank Eric von Elert and Nelson Hairston for their useful comments on the manuscript. This study is part of the EVENET network.

## Author contributions

E.M., M.C., L.D.M. and E.D. designed the experiment. E.M. and M.C. performed the experiment and analyzed the data. E.M. wrote the first draft of the manuscript. All authors participated in discussions and editing of the manuscript.

## Additional information

**Competing interests:** The authors declare no competing financial interests.

