## [Peer Review File · Nature Communications]

Reviewers' comments:

Reviewer #1 (Remarks to the Author):

The investigation of gut microbiological communities and their effects on fitness, physiology and protection against pathogens of the respective host is currently a hot topic in environmental medicine and ecology. Here the authors address the question in how far different gut communities determine the tolerance of the host *Daphnia* to a toxic food item, which is a bacterium itself (cyanobacterium). Four genotypes of *Daphnia*, that differ in their tolerance of a cyanobacterium, were grown for several months in the absence/presence of this cyanobacterium with regular additions of natural surface water in order to provide a somewhat natural bacterial community. Then these animals ('donors') were used for the preparation of gut homogenates that were subsequently transferred (transplant experiment) to the same genotypes (recipient genotypes). Donors with a high tolerance resulted in elevated tolerance of the receiver, and this effect was stronger when donors had been fed cyanobacteria; sensitive donors lead to less tolerant recipient genotypes. These are impressively strong effects, and they are new with respect to interactions of *Daphnia* and cyanobacteria. The experimental design with replicated lines of preconditioning and subsequent replicated transplantations is adequate to assess the effects of transplants on fitness. However, the analysis of the microbiomes is unfortunately less consistent. I have listed crucial aspects below.

Major points

-the 'transplant experiments' show impressively strong effects on the tolerance of the recipient genotypes. However, it has not been tested that bacteria have indeed been transplanted, i.e., there are no data on the microbiomes of the recipient genotypes. Although there is strong circumstantial evidence that the observed effects may have been caused by transmitted bacteria, this mechanism, which is crucial for the overall message of the paper, has not been demonstrated.

-the analysis of the bacterial communities of the donors has not been replicated. Therefore it cannot be assessed if (i) the microbiomes of the different *Daphnia* genotypes x food treatments differed among lines and treatments. More precisely it cannot be tested to which degree the microbiomes of tolerant and sensitive *Daphnia* lines differed and if microbiomes were more similar among similarly tolerant lines. Statements like 'prior exposure to *Microcystis* alters the structure of the gut microbial community' (line 106) are not supported by the analyses.

- Although not replicated (see above) the microbiome data tentatively suggest that, after several months of exposure to the same food conditions, the different *Daphnia* genotypes have acquired different microbiota (genotype effect on microbiome acquisition). In the subsequent 'transplant experiment' the recipient genotypes were exposed to highly concentrated foreign gut homogenates for two days and were then monitored for two weeks. I wonder if the strong effects of 'transplantations' that were observed here are persistent: the above mentioned genotype effect on microbiome acquisition suggests that the putatively transplanted bacterial community will not be stable within the new host. This, however, suggests that the observed effects of transplantation are not persistent, which does not support the idea that they are ecologically or evolutionary relevant.

Minor points

I understand that the interaction of *Daphnia* and cyanobacteria is such that cyanobacteria have negative effects on survival and reproduction of *Daphnia*. In Extended Data Figure 1 genotype R1 is showing a 40% better performance on the cyanobacterium. I wonder how representative this performance is for the *Daphnia* x cyanobacteria interaction, given that this genotype is one out of four that were experimentally investigated.

In Extended Data Figure 1 genotype S2 has a mortality of almost 60% within 12 days under control conditions (green alga only). I wonder how it can happen that this genotype was not lost during the six months of exposure to the green alga only.

The authors repeatedly argue against a direct role of *Daphnia* genotype on tolerance to the cyanobacterium. Although I understand the enthusiasm of the authors, I think this is too excessive: The fact that all genotypes that have been exposed to the microbiome of R2 (Fig. 2 c)

show a lower survival than R2 itself (Fig. 2 a) points at the genotype effects of R2 (probably one could even quantify this genotype effect by calculating the difference). The same holds for R1. In line with this, I do not share the reasoning (line 64) that the finding that 'all recipient genotypes had similar survival and reproduction probability upon cyanobacterial exposure' suggests 'that gut microbiota rather than Daphnia genotype itself drives resistance'. This is not conclusively logic, instead the similar values may rather be accidental.

line 98: 'the mechanisms underlying adaptation and acclimatization remained largely unknown'. I have had a look at the Abstracts of ref. 23-24 and do not understand what is meant by 'largely unknown', please explain.

lines 98 and 147: I do not understand in how far the results obtained here may explain genetic responses or adaptation to cyanobacteria. How can the genome of the host be affected by its symbiotic bacterial community?

The term 'toxic' or 'harmful' seems to be due to effects of cyanobacteria on warm blooded animals, but it remains unclear, which metabolites are involved and in how far this translates into effects on Daphnia. I furthermore suggest to replace 'resistant' by 'tolerant', as the system investigated here is not a host-pathogen system.

Fig. 2 b): where is genotype S2?

Fig. 3: legend: replace 'c' by 'b' and 'd' by 'c'.

Extended Data Figure 1: explain the blue and red line.

Reviewer #2 (Remarks to the Author):

Review of "Genotype-dependent gut microbiota drives zooplankton resistance to toxic cyanobacteria"

By E. Make et al.

Reviewed by Nelson Hairston, Jr. (signed review)

This is an exceptionally interesting and original study: I believe it is the first of its kind. The authors show convincingly that the gut microbiota of Daphnia are critical in determining the sensitivity of the animals to dietary cyanobacteria. As the authors point out, there is now abundant evidence, for a variety of kinds of animals, that the composition of the gut microbiome plays a crucial role in the physiology and health of the host in a way that is diet-dependent. What sets this study apart, however, is the ecological significance of the interaction the authors investigated. Daphnia is the major phytoplankton grazer in many lakes, and Microcystis is an important nuisance bloom-forming cyanobacterium, known to be poor food for Daphnia. This discovery that community composition of bacteria in the gut has a strong effect on the ability of the grazers to cope with what is typically lousy food has the potential to transform our understanding of zooplankton-phytoplankton dynamics in lakes.

The methods are generally robust as is the interpretation of the results, though others will have to comment on the details of the metagenomic analyses.

My only substantial reservation about the paper as written is that the authors draw a more general conclusion than their experiment can support about the lack of importance of genetic variation among Daphnia lineages in their sensitivity to dietary cyanobacteria. For example, in the first paragraph of the text where they begin the general discussion of their findings, they write: "Our results demonstrate that Daphnia genotype by itself does not have a strong effect on resistance to cyanobacteria ..." (lines 98-101). This is certainly true for the four clones of Daphnia magna that they studied, but it seems likely that the reason that this is the case is that the clones they happened to have chosen had little genetic variation for cyanobacterial resistance. The lineages

came from three different water bodies, but the authors give little information about the food environment where they lived and so whether natural selection might have differed to any great extent. They do point out that one of the lakes was oligotrophic while another one sometimes has *Microcystis* present, but this isn't much to go on. If genetic variation for sensitivity to dietary cyanobacteria happened to be low in the lineages they chose, then this would have strongly biased their results against finding an effect of genotype.

I am signing this review because the authors cite (favorably) a study by my colleagues and me on evolution of resistance to dietary cyanobacteria in a population of *Daphnia galeata*, and I want to use this example as a situation where differences in gut microbiota cannot explain the strong clonal differences in resistance we observed. Rather, the differences must have been genetic. Like the *Daphnia* clones the authors used, ours originated from the hatching of diapausing eggs. These eggs were produced at different times in the history of the study lake before and after cultural eutrophication when cyanobacteria were either essentially absent, or had been abundant for a decade. In contrast to the *D. magna* used by the authors of the present study, the 32 *D. galeata* clones in our investigation had experienced strongly different natural selection regimes for extended periods. I agree with the authors that offspring hatching from diapausing eggs must have left the gut microbiota of their mother behind (lines 373-374). And, because in our study the *D. galeata* were reared after hatching all on the same good food diet (for > 40 generations), there was no reason for distinct gut microbiome communities to develop in the different clones. Finally, our assay of clonal response to dietary cyanobacteria (*Microcystis*, just as the authors used in their study) was a measure of somatic growth rate over a 6 day period from immature to adult, so there was very little opportunity for different microbiota communities to develop, and no difference in how the *Daphnia* clones were treated. The marked differences in tolerance to dietary cyanobacteria had to have been genetically based because the clones were known to be genetically distinct. All of this is just to say that the authors need to moderate their enthusiasm for their results - fascinating as the findings are - and allow in general for both *Daphnia* genotype and gut microbiota to be important. Our experiment on *D. galeata* tested only for genetic differences, and the present study tested only the effects of gut microbiota. The next step will be to take *Daphnia* that have experienced natural selection for many generations in good food or poor food (cyanobacteria rich) environments and then compare genetic effects with the importance of gut microbiota. Again, none of this is to say that the present study is anything but fascinating, exciting, and novel. It is all of those things. It just isn't as general as the discussion implies.

The authors may want to cite the excellent study by Jiang et al. (2015. *Microgeographic adaptation to toxic cyanobacteria in two aquatic grazers. Limnology and Oceanography* 60:947-956) which explores local adaptation in *Daphnia* populations from lakes with high and low cyanobacterial densities. I would have said that this study also showed genetic adaptation in the grazers, but because those investigators collected their animals directly from the water column and only reared them in the lab through three generations before testing for response to diet, it is possible, in light of the present study, that the differences observed were at least in part the product of differences in maternally transmitted gut microbiota. However, even in this case, it would be necessary to address the fact that all clones in that study were reared for those three generations on only good food before being experimentally exposed of good and poor food treatments of a single generation.

Two more minor questions whose answers would strengthen the paper are: 1. Do the authors know that the strain of *Microcystis aeruginosa* they used was actually toxic? Many strains of this species have high concentrations of the toxin microcystin, but other strains, readily obtained from culture collections, lack the toxin. We need to know which it is for this study, especially since the authors appear to assume in the discussion that the strain is toxic and not just nutritionally poor quality (i.e., lacking essential fatty acids). And 2. Did the authors do a metagenomic scan of the guts of the "germ-free" *Daphnia* to see what if any microbiota were present before feeding started? Perhaps the latter wasn't possible since, I suppose, it would have to have been carried out on juveniles. Still it would be very interesting to see how the microbiota composition

developed over time.

Finally, there are other questions worth considering for this system. What benefit do the microbiota get from making the *Daphnia* less sensitive to cyanobacteria in their diet. Do these microbes benefit directly because they are also making the cyanobacteria more usable for themselves? Does the *Daphnia* make the cyanobacteria more available to the microbes? Does a healthy *Daphnia* make life better for its microbes in some other way? Is there a cost to the microbes for making the cyanobacteria more usable so that in the absence of cyanobacteria a different group of microbes have an advantage? I don't suggest that the authors should have answers to such questions, but it would be good for them to indicate to their readers that they recognize that such questions are critical to answer at some point.

Very specific, small editing comments.

Line(s)

18 "... threats to livestock..."

49 Nowhere in the main body of the paper are we told what species of *Daphnia* is used. That information, currently only found in the Methods, should be mentioned up front.

97 "... genes that respond to ..."

150 "... our study improves our understanding..."

152-156 I think these two sentences are getting too far from the main topic. I suggest omitting them.

Fig. 1 I've always found smiley faces rather annoying, so their presence here seems too cute to me.

But that's just my own bias.

All figs. Using red and green to contrast treatments will be problematic for people who are red-green

Color blind. I suggest choosing a different color combination.

177;1 88 "... on survival following exposure to toxic..." [Also, here is an example of calling the *Microcystis* toxic without evidence having been presented.]

Fig. 3 put the stats results on (a) and (c) even if they are ns.

190-191 (c) and (d) should be (b) and (c)

Fig. 4 Seems like the information in panels (a) and (b) are a bit redundant. I think (b) could be moved

to the on line content. Also, the grey circles (not "spheres") in (b) are pretty difficult to see.

I also don't think the rarefaction curves need to be presented here. The comparison can be given in the text and the figure put in the on line content.

401 "genotypes"

406 "... (each containing 60 ..."

406-407 "... jar was fed a green..."

412 "...on the condition of the ..."

415 "... kept the same among all jars."

420 "... were subjected to these diets for six months."

425 "... deionized sterile water..."

452 "For this purpose..."

558 "... were chosen as ..."

Reviewer #3 (Remarks to the Author):

In this study, the authors investigated host protective effects of gut microbiota against toxic cyanobacteria, which is a timely and exciting topic. By using *Daphnia* as an experimental model, they report that resistance to toxic cyanobacteria in host is driven by the gut microbiota.

Recipients that were inoculated with gut extracts from resistant donor genotypes, represent increased resistance to toxic cyanobacteria. They also provide evidence that a pre-exposure of donors to toxic cyanobacteria for 6 months resulted in enhanced protective effect of the microbiota against feeding toxic cyanobacteria (or cyanobacterial toxins). Finally, they report evidence that host genotype and diet interact to shape both structure and functionality of microbiota. Based on metagenomics analysis of gut microbiota from dissected guts of donors, they represent different bacterial communities in response to host genotypes (susceptible and resistance to toxic cyanobacteria) and diets (non-toxic green algae and toxic cyanobacteria). While this study sounds initially appealing, I find it unsatisfactory and preliminary since several major conclusions are partly supported by the reported observations and several important controls in experiments are missing, often leading important conclusions that are likely not true.

Please find below a list of major issues and claims made in the manuscript that I think, which are not supported by the current data, and would definitively be needed to either be strengthened or fully re-addressed for the story to be conclusive.

1. *Daphnia* genotype affects the level of resistance to toxic cyanobacteria by modulation of their gut microbiota:

In animals, transgenerational effects contain not only the vertical transmission of microbiota and epigenetic modulations which are extensively discussed in this manuscript, but also genetic variation (e.g., SNP) and immune priming. I am not insisting that the authors should evaluate SNP or immune priming of host animal. However, the authors at least provide how the levels of systemic and local immune responses in host *Daphnia* differ in response to feeding cyanobacteria. Immune tolerance as well as immune resistance is important response against invading pathogens. Thus, the authors must verify a disparity in immune responses against toxic cyanobacteria between: (i) susceptible and resistance donor genotypes; (ii) donor pre-exposed to a non-toxic green algae diet and donor pre-exposed to a toxic cyanobacteria diet; (iii) donor and recipient. This experiment will enable us to better understand not only the cyanobacteria-mediated host immune modulation that may contribute to shape the resistant phenotype of gut microbiota against co-existing pathogens, but also the strategy in *Daphnia*'s innate immunity (immune tolerance or resistance) against toxic cyanobacteria.

2. Microbiota from resistant genotypes conferred a higher resistance to recipient *Daphnia* than microbiota from susceptible genotype:

First, this reviewer could not find any evidence for transmission of the microbiota from donors to recipients. It would be so much helpful if the authors provide metagenomics data of recipient gut microbiota. In addition to culture-independent data (e.g., metagenomic analysis), they must provide evidence for successful colonization and/or temporal existence of the transplanted microbiota in the gut of germ-free recipients (i.e., culture-dependent experiments such as CFU test). Most importantly, to fully support the conclusion, they must reveal the candidate microbe(s) mostly responsible for the resistance against toxic cyanobacteria at the species level. According to Figure 4 and Extended data Figure 3, it is expected that microbes in the gut of *Daphnia* are mostly aerobic. Thus, this reviewer strongly recommends the authors isolate bacterial species (e.g., *Pseudomonas* sp. found from the resistant donor, as in the Extended data Figure 3) from the gut of *Daphnia*, and then evaluate the resistance of the gnotobiotic animals harboring the candidate bacterial species to toxic cyanobacteria. Identifying the candidate bacterial species is particularly important because *Daphnia* might reflect a high spatiotemporal variation of bacterial community in their gut. For example, the structure and composition of *Daphnia* gut microbiota are differed among different donor genotypes (Figure 3), and between this study and previous study reported by Decaestecker's lab. (Gallens M et al., 2015 Food availability affects the strength of mutualistic host-microbiota interactions in *Daphnia magna*. ISME J), suggesting absence of core microbiota in the gut of *Daphnia*. Therefore, the different microbial community observed in cyanobacteria pre-exposed group possibly merely reflects spatiotemporal changes in bacterial communities. Considering these issues, claiming additional experiments such as bacterial isolation and mono-association tests are not an unreasonable demand.

3. The second paragraph in main text reporting that gut microbiota rather than Daphnia genotype itself drives resistance to toxic cyanobacteria is confusing. Germ-free control should be included in Figure 2b, c and d, and Figure 3.

4. In Figure 4, I see only a single biological replicate for the metagenomics analysis. More biological replicates (at least n=3) need to be implemented for the sake of the statistical analysis robustness. Also, the number of PCR cycle (35 cycles) is not low. Add technical replicate (at least n=3) to minimize a potential PCR bias.

5. In Figure 4d, Daphnia pre-exposed to a toxic cyanobacteria diet represent higher values of microbial richness and evenness than Daphnia pre-exposed to a non-toxic green algae diet. Provide reasonable discussion about potential factor(s) influencing the microbial diversity in the gut of Daphnia.

Specific comments:

- Line 87. Correct spacing error.

- Figure 3 legends. Correct the order. It should be a, b and c.

Response to Reviewer #1:

The investigation of gut microbiological communities and their effects on fitness, physiology and protection against pathogens of the respective host is currently a hot topic in environmental medicine and ecology. Here the authors address the question in how far different gut communities determine the tolerance of the host Daphnia to a toxic food item, which is a bacterium itself (cyanobacterium). Four genotypes of Daphnia, that differ in their tolerance of a cyanobacterium, were grown for several months in the absence/presence of this cyanobacterium with regular additions of natural surface water in order to provide a somewhat natural bacterial community. Then these animals ('donors') were used for the preparation of gut homogenates that were subsequently transferred (transplant experiment) to the same genotypes (recipient genotypes). Donors with a high tolerance resulted in elevated tolerance of the receiver, and this effect was stronger when donors had been fed cyanobacteria; sensitive donors lead to less tolerant recipient genotypes. These are impressively strong effects, and they are new with respect to interactions of Daphnia and cyanobacteria. The experimental design with replicated lines of preconditioning and subsequent

replicated transplantations is adequate to assess the effects of transplants on fitness. However, the analysis of the microbiomes is unfortunately less consistent. I have listed crucial aspects below.

We thank the reviewer for his/her positive and constructive comments. For the extra analyses on the microbiome we refer to the comments below.

Major points

-the 'transplant experiments' show impressively strong effects on the tolerance of the recipient genotypes. However, it has not been tested that bacteria have indeed been transplanted, i.e., there are no data on the microbiomes of the recipient genotypes. Although there is strong circumstantial evidence that the observed effects may have been caused by transmitted bacteria, this mechanism, which is crucial for the overall message of the paper, has not been demonstrated.

In the revised version of the manuscript, we characterized the gut microbiota composition in all replicated donor populations (4 genotypes * 2 diets * 3 replicates; Experiment 2), which provided us with all the information necessary to determine the effects of both host genotype (i.e. tolerant *versus* susceptible) and diet on the *Daphnia* gut microbiota (see methods lines 610-673 and results lines 165-198 + Figures 4 and Supplementary Figure 3). This experiment showed a clear difference in gut microbiota composition between tolerant and susceptible genotypes. Together with the phenotypic effects observed in the transplant experiment, these results indicate a role of gut symbionts in tolerance to cyanobacteria.

In addition, we now performed a new transplant experiment (Experiment 3), in which we inoculated different genotypes with different microbial inocula, and then monitored the gut microbiota composition of recipient *Daphnia* over time (1, 7 and 14 days after the transplant; see methods lines 675-752 and results lines 200-249 + Figure 5). This new transplant experiment (1) confirmed that gut microbes successfully establish in the gut after a transplant (which had already been demonstrated in previous studies; see Callens et al. 2016), (2) determined the relative importance of the donor *versus* recipient genotype effect, and (3) documented the dynamics of the gut microbial community as the *Daphnia* age, establishing that the donor effect is persistent over time. For this new transplant, the genotypes were chosen among the *Daphnia* clones available in the laboratory at the moment of the experiment. As some of the original clonal cultures had been discarded, this experiment was done on clones different from those used in Experiments 1 and 2. Yet, this experiment clearly shows both a donor and a recipient genotype effect on the microbiota composition, and provides insight on the lasting effect of donor microbiota.

Combining the results of Experiment 2 and 3 provides a strong account on (a) the gut microbiota composition of *Daphnia* in function of both their genotype and diet, (b) the impact of donor microbiota on gut microbiota composition in recipients, (c) the dynamics of the gut microbiota composition over time, including an assessment of the relative contribution of donor and recipient genotype.

*-the analysis of the bacterial communities of the donors has not been replicated. Therefore, it cannot be assessed if (i) the microbiomes of the different *Daphnia* genotypes x food treatments differed among lines and treatments. More precisely it cannot be tested to which degree the microbiomes of tolerant and sensitive *Daphnia* lines differed and if microbiomes were more similar among similarly tolerant lines. Statements like 'prior exposure to *Microcystis* alters the structure of the gut microbial community' (line 106) are not supported by the analyses.*

The characterization of the gut microbiota in the donor populations has been performed again, now fully replicated with 3 replicates per genotype and per diet (see our response above). Hence, we could perform appropriate analyses to compare microbiota among tolerant and susceptible genotypes, and among diets. These results were integrated in the revised version of the manuscript (lines 165-198 + Figure 4).

- Although not replicated (see above) the microbiome data tentatively suggest that, after several months of exposure to the same food conditions, the different *Daphnia* genotypes have acquired different microbiota (genotype effect on microbiome acquisition). In the subsequent 'transplant experiment' the recipient genotypes were exposed to highly concentrated foreign gut homogenates for two days and were then monitored for two weeks. I wonder if the strong effects of 'transplantations' that were observed here are persistent: the above mentioned genotype effect on microbiome acquisition suggests that the putatively transplanted bacterial community will not be stable within the new host. This, however, suggests that the observed effects of transplantation are not persistent, which does not support the idea that they are ecologically or evolutionary relevant.

The referee is right that there is a “tension zone” between a transplant effect (source microbiota conferring tolerance to cyanobacteria) and the host genotype effect. It is likely that indeed there is a source microbiota effect that is gradually influenced by sorting mediated by host genotype. To obtain insight in this, we carried out an additional transplant experiment (Experiment 3 in the revised version of the manuscript) in which we monitored gut microbiota composition in recipient *Daphnia* at different time points after the transplant (1, 7 and 14 days). Our results revealed on the one hand a strong effect of time on the taxonomic composition of the gut microbiota (maybe in response to temporal variation in metabolic demands), but also both a persistent donor microbiota effect as well as a growing influence of host genotype through time. The relative contribution of the inoculum and of the recipient genotype to the gut microbiota structure indeed varied over time. Soon after the transplant (1 day), variation in gut microbiota composition was mainly explained by the inoculum effect (i.e. donor effect), while the recipient genotype did not have an effect. Two weeks after the transplant, variation in the gut microbiota composition was explained by both the inoculum and the recipient genotype. Together, these results suggest that the exogenous microbial exposure (i.e. donor effect) determines the establishment and the initial structure of the gut microbiota in *Daphnia*. Although this inoculum effect is persistent (it is at least still observed after two weeks), the structure of the gut microbiota is progressively reshaped in later life stages, and starts to diverge between recipient genotypes. The latter suggests the occurrence of host genotype mediated internal sorting processes. This information has been integrated in the revised version of the manuscript (see results lines 200-249 + Figure 5).

Hence, our new results show that the donor effects persist over time, even if the structure of the gut microbiota is progressively reshaped. The results of Experiment 3 also confirm that the gut microbiota composition is to a significant degree shaped by host genotype. We want to stress that in our study, gut microbiota transplants are a tool to assess the role of the gut microbiota in *Daphnia* tolerance to cyanobacteria. We show that a susceptible genotype inoculated with the microbiota from a tolerant clone becomes tolerant, and *vice-versa*, thus pointing out the microbiota as a major driver of cyanobacterial tolerance. Our results do not only imply that gut microbiota are important in providing tolerance to cyanobacteria, but also imply that the gut microbiota are to some degree impacted by host genotype. In other words, observations on inter-clonal variation in *Daphnia* tolerance to cyanobacteria can to some extent be explained by genetic variation in microbiota composition, which is relevant from an ecological and an evolutionary point of view. Indeed, when facing cyanobacterial blooms, *Daphnia* harboring a set of microbes that confer them a better tolerance to cyanobacteria are expected to be selected, resulting in a genetic adaptation of the host population towards harboring genes that steer the microbiota in the direction to confer tolerance. Selection may, for example, occur on immune genes, which are known to be involved in the structuration of the gut microbiota (see lines 49-52, 271-273, 386-399).

Minor points

I understand that the interaction of *Daphnia* and cyanobacteria is such that cyanobacteria have negative effects on survival and reproduction of *Daphnia*. In Extended Data Figure 1 genotype R1 is showing a 40% better performance on the cyanobacterium. I wonder how representative this performance is for the *Daphnia* x cyanobacteria interaction, given that this genotype is one out of four that were experimentally investigated.

Several earlier studies (e.g. Lemaire et al. 2012, Jiang et al. 2013) similarly found an important amount of genetic variation in *Daphnia* tolerance to cyanobacteria. Jiang et al. (2013) found that thirteen sympatric *Daphnia pulex* clones differed significantly in fitness when fed *Chlorella pyrenoidosa* and toxic *Microcystis aeruginosa*. The least susceptible

clone was 12 times less susceptible to *M. aeruginosa* than the most susceptible clone. Similar to our study, they found that for some genotypes survival was either non-affected, or slightly increased, when fed toxic cyanobacteria. Such intraspecific variation among contemporaneous clones suggests a high potential for a co-evolutionary arms race between cyanobacteria and grazers. This issue is now discussed in lines 261-267 of the revised version of the manuscript.

In Extended Data Figure 1 genotype S2 has a mortality of almost 60% within 12 days under control conditions (green alga only). I wonder how it can happen that this genotype was not lost during the six months of exposure to the green alga only.

The key observation to explain this is that in general, *Daphnia* perform much better when filtered pond water is added to the medium, probably due to the presence of beneficial symbionts or additional nutrients/vitamins. When we assessed cyanobacterial tolerance in our *Daphnia* genotypes (Figure 1 of the revised manuscript), we did not add pond water to avoid any confounding factor and achieve the highest possible level of standardization. This likely influenced the fitness of the *Daphnia*. In our experimental donor populations (exposed to either toxic or non-toxic food), pond water was regularly added to provide the animals with a diverse pool of symbionts, hence the populations performed quite well.

The authors repeatedly argue against a direct role of Daphnia genotype on tolerance to the cyanobacterium. Although I understand the enthusiasm of the authors, I think this is too excessive: The fact that all genotypes that have been exposed to the microbiome of R2 (Fig. 2 c) show a lower survival than R2 itself (Fig. 2 a) points at the genotype effects of R2 (probably one could even quantify this genotype effect by calculating the difference). The same holds for R1. In line with this, I do not share the reasoning (line 64) that the finding that 'all recipient genotypes had similar survival and reproduction probability upon cyanobacterial exposure' suggests 'that gut microbiota rather than Daphnia genotype itself drives resistance'. This is not conclusively logic, instead the similar values may rather be accidental.

We agree with the reviewer that claiming that *Daphnia* genotype does not have any direct effect on resistance to cyanobacteria is too excessive. We actually did not intend to write it in such strong terms; rather we wanted to stress that in our experiment much of the variation can be explained by the microbiome. We now removed such sentences from the manuscript. In the revised version of the manuscript, we now also added the results of Experiment 3, which provide a perspective on how *Daphnia* genotype can impact microbiome composition and in this way play a role in conferring tolerance to cyanobacteria. In addition, there are some studies (e.g. Schwartzenberger et al. 2014) that clearly indicate that there might be some direct effects of *Daphnia* genotype on tolerance to cyanobacteria (see lines 319-322). We now try to formulate this better in the revised version of the manuscript.

Our transplant experiment revealed that when all recipient genotypes are inoculated with a same microbial inoculum, all genotypes exhibit similar survival upon cyanobacterial exposure. Hence, differences in fitness between tolerant and susceptible clones can no longer be observed when they have been transplanted with a same microbial inoculum. We do not think that such result can occur by chance, given the strong differences that were initially observed between tolerant and susceptible clones. In addition, strong differences were observed between recipient *Daphnia* that received different microbial inocula (i.e. significant donor effect). The fact that cyanobacterial tolerance in recipient *Daphnia* only depends on the donor, and not on the recipient genotype itself, thus points out the gut microbiota as an important mediator of cyanobacterial tolerance, and suggests that genetic variation in cyanobacterial tolerance is mainly due to genetic variation in the gut microbiota profile. The host genotype thus indeed affects cyanobacterial tolerance, but our results suggest that it mainly happens through the gut microbiota. This is now discussed in a more elaborated way (lines 261-276).

line 98: 'the mechanisms underlying adaptation and acclimatization remained largely unknown'. I have had a look at the Abstracts of ref. 23-24 and do not understand what is meant by 'largely unknown', please explain.

This sentence has now been removed. What we meant by “largely unknown” is that the precise mechanisms, e.g. the role of the microbiome, has not been investigated. We now reformulated the statement, however.

lines 98 and 147: I do not understand in how far the results obtained here may explain genetic responses or adaptation to cyanobacteria. How can the genome of the host be affected by its symbiotic bacterial community?

Our study reveals that gut microbiota in *Daphnia* is genotype-dependent, and that genetic variation in the composition of the gut microbiota are responsible for genetic variation in *Daphnia* tolerance to cyanobacteria. These results strongly suggest that tolerant genotypes are tolerant, at least partly, because they have “tolerant” microbiota, and also that the composition of the microbiota is at least partly mediated by host genotype. These tolerant genotypes are expected to be selected when facing cyanobacterial blooms, resulting in genetic adaptation of the population.

We do not think that the microbiota can affect the host genome. Instead, we think that the host genome shapes the gut microbiota, likely through the intercession of the immune system. It is indeed known that immune components, such as antimicrobial peptides produced in the digestive tract, play an important role in the structuring of the gut microbiota (see e.g. Tasiemski et al. 2015 and Ostaff et al. 2013). Even if the gut microbiota is acquired independently each generation from environmental bacteria or horizontal transfer, the host genetic background might have an effect on the recruitment of these bacteria, and on their persistence in the digestive tract. In that sense, gut symbionts are considered to be, at least partly, heritable (see Shapira 2016 and Goodrich et al. 2014). Hence, the gut microbiota may be seen as an extended phenotype, on which selection can act. Our idea is that host genes involved in the structuring of the gut microbiota (e.g. immune genes) would evolve upon cyanobacterial exposure, resulting in an evolution of the microbiota structure, and thus an evolution of the *Daphnia* tolerance.

This points are now explained in the revised version of the manuscript (lines 49-52, 271-273, 386-399).

The term 'toxic' or 'harmful' seems to be due to effects of cyanobacteria on warm blooded animals, but it remains unclear, which metabolites are involved and in how far this translates into effects on Daphnia. I furthermore suggest to replace 'resistant' by 'tolerant', as the system investigated here is not a host-pathogen system.

The mechanisms underlying cyanobacterial toxicity in invertebrates like *Daphnia* are indeed not precisely known. However, it has been demonstrated that microcystin (one of the toxins produced by *Microcystis*) was detrimental to *Daphnia*, and there is a rich literature on negative effects of cyanobacteria on *Daphnia* (e.g. Hairston et al. 2001; Rohrlack et al. 2001; Sarnelle et al. 2010; Schwarzenberger et al. 2014). *Daphnia* survival was lower when exposed to a *Microcystis* strain that produces microcystin than when exposed to the mutant version of this *Microcystis* strain, that does not produce the toxin (see e.g. Schwarzenberger et al. 2014). In addition, metabolites other than microcystin, such as protease inhibitors, were also found to affect *Daphnia* survival. More information is now provided on the toxic effects of *Microcystis* (lines 476-482).

We replaced “resistant” by “tolerant” everywhere in the manuscript.

Fig. 2 b): where is genotype S2? In the initial version of the manuscript, we did not have data for the sample corresponding to genotype S2 (we did not obtain enough material for the sequencing). The whole experiment has, however, been performed again, and the revised version now includes replicated data for all genotypes.

Fig. 3: legend: replace 'c' by 'b' and 'd' by 'c'. The figure has changed, as well as the associated legend.

Extended Data Figure 1: explain the blue and red line. Performed accordingly.

Response to Reviewer #2:

Review of "Genotype-dependent gut microbiota drives zooplankton resistance to toxic cyanobacteria"

By E. Make et al.

Reviewed by Nelson Hairston, Jr. (signed review)

This is an exceptionally interesting and original study: I believe it is the first of its kind. The authors show convincingly that the gut microbiota of Daphnia are critical in determining the sensitivity of the animals to dietary cyanobacteria. As the authors point out, there is now abundant evidence, for a variety of kinds of animals, that the composition of the gut microbiome plays a crucial role in the physiology and health of the host in a way that is diet-dependent. What sets this study apart, however, is the ecological significance of the interaction the authors investigated. Daphnia is the major phytoplankton grazer in many lakes, and Microcystis is an important nuisance bloom-forming cyanobacterium, known to be poor food for Daphnia. This discovery that community composition of bacteria in the gut has a strong effect on the ability of the grazers to cope with what is typically lousy food has the potential to transform our understanding of zooplankton-phytoplankton dynamics in lakes.

The methods are generally robust as is the interpretation of the results, though others will have to comment on the details of the metagenomic analyses.

We thank Prof. Hairston for his positive comments.

My only substantial reservation about the paper as written is that the authors draw a more general conclusion than their experiment can support about the lack of importance of genetic variation among Daphnia lineages in their sensitivity to dietary cyanobacteria. For example, in the first paragraph of the text where they begin the general discussion of their findings, they write: "Our results demonstrate that Daphnia genotype by itself does not have a strong effect on resistance to cyanobacteria ..." (lines 98-101). This is certainly true for the four clones of Daphnia magna that they studied, but it seems likely that the reason that this is the case is that the clones they happened to have chosen had little genetic variation for cyanobacterial resistance. The lineages came from three different water bodies, but the authors give little information about the food environment where they lived and so whether natural selection might have differed to any great extent. They do point out that one of the lakes was oligotrophic while another one sometimes has Microcystis present, but this isn't much to go on. If genetic variation for sensitivity to dietary cyanobacteria happened to be low in the lineages they chose, then this would have strongly biased their results against finding an effect of genotype.

We agree with Prof. Hairston that claiming that *Daphnia* genotype does not have any effect on resistance to cyanobacteria is too excessive, and poorly formulated. We now removed such sentences from the manuscript and provided more context and a more nuanced vision.

In our study, when testing for cyanobacterial resistance among clones, we actually found some genetic variation, with 2 tolerant and 2 susceptible clones. Hence, there is a genotype effect. We now clearly mention it, specifying that our study corroborates previous results (lines 261-267). However, our transplant experiment revealed that when all recipient genotypes were inoculated with a same microbial inoculum, all genotypes exhibited similar survival upon cyanobacterial exposure. Differences between tolerant and susceptible clones could thus no longer be observed when they had been transplanted with a same microbial inoculum. In addition, strong differences were observed between recipient *Daphnia* that received different microbial inocula (i.e. significant donor effect). The fact that cyanobacterial tolerance in recipient *Daphnia* only depends on the donor, and not on the recipient genotype itself, thus points out

the gut microbiota as an important mediator of cyanobacterial tolerance, and suggests that genetic variation in cyanobacterial tolerance in this set of clones is mainly due to genetic variation in the gut microbiota profile. The host genotype thus indeed affects cyanobacterial tolerance, but our results show that this mainly happens through the gut microbiota. We now explain this better in the revised version of the manuscript (lines 267-276). Overall, we see two levels at which host genotype can play a role. First, several studies reported genetic differences in gene expression among *Daphnia* clones that are differentially susceptible to cyanobacteria (e.g. Schwarzenberger et al. 2014). Secondly, *Daphnia* genotypes might be impacting the structure of the gut microbiota and in this way contribute to genotypic differences in cyanobacteria tolerance.

As mentioned to reviewer 1, the host genetic background likely has an effect on the recruitment of bacterial symbionts, and on their persistence in the digestive tract. This may for example occur through the intercession of the immune system (see e.g. Tasiemski et al. 2015). Hence, the gut microbiota may be seen as an extended phenotype, on which selection can act. Our hypothesis is that the host's genes involved in the structuration of the gut microbiota (e.g. immune genes) would evolve upon cyanobacterial exposure, resulting in an evolution of the microbiota structure, and thus an evolution of the *Daphnia* tolerance. We now discuss this in the revised version of the manuscript (lines 49-52, 271-273, 386-399).

It is true that we do not have much information on the *Microcystis* history of ponds which our experimental clones come from. Hence, we cannot draw precise conclusion on the process of adaptation to cyanobacteria in these populations (in any way such conclusions would not be very strong and reliable given that we worked with only four clones). However, our results provide us with enough information to prove that (1) there is genetic variation in *Daphnia* tolerance to cyanobacteria (consistent with previous studies performed on *Daphnia*, including Hairston et al. 1999 and Hairston et al. 2001) and that (2) genetic variation in resistance to cyanobacteria relies, at least partly, on genetic variation in the gut microbiota composition. Together, these results point out the gut microbiota as a driver of genetic adaptation to cyanobacteria.

I am signing this review because the authors cite (favorably) a study by my colleagues and me on evolution of resistance to dietary cyanobacteria in a population of Daphnia galeata, and I want to use this example as a situation where differences in gut microbiota cannot explain the strong clonal differences in resistance we observed. Rather, the differences must have been genetic. Like the Daphnia clones the authors used, ours originated from the hatching of diapausing eggs. These eggs were produced at different times in the history of the study lake before and after cultural eutrophication when cyanobacteria were either essentially absent, or had been abundant for a decade. In contrast to the D. magna used by the authors of the present study, the 32 D. galeata clones in our investigation had experienced strongly different natural selection regimes for extended periods. I agree with the authors that offspring hatching from diapausing eggs must have left the gut microbiota of their mother behind (lines 373-374). And, because in our study the D. galeata were reared after hatching all on the same good food diet (for > 40 generations), there was no reason for distinct gut microbiome communities to develop in the different clones. Finally, our assay of clonal response to dietary cyanobacteria (Microcystis, just as the authors used in their study) was a measure of somatic growth rate over a 6 day period from immature to adult, so there was very little opportunity for different microbiota communities to develop, and no difference in how the Daphnia clones were treated. The marked differences in tolerance to dietary cyanobacteria had to have been genetically based because the clones were known to be genetically distinct. All of this is just to say that the authors need to moderate their enthusiasm for their results - fascinating as the findings are - and allow in general for both Daphnia genotype and gut microbiota to be important. Our experiment on D. galeata tested only for genetic differences, and the present study tested only the effects of gut microbiota. The next step will be to take Daphnia that have experienced natural selection for many generations in good food or poor food (cyanobacteria rich) environments and then compare genetic effects with the importance of gut microbiota. Again, none of this is to say that the present study is anything but fascinating, exciting, and novel. It is all of those things. It just isn't as general as the discussion implies.

We fully agree with Prof. Hairston (see also our answers to some other comments) and now moderated our conclusions in the revised version of the manuscript, and mentioned that there is genetic variation in cyanobacterial tolerance (lines 261-267). We, however, would also argue that our results do not contradict previous studies showing

genetic differences in resistance to cyanobacteria or genetic adaptation (see our comment above). In addition, our additional experiment studying the dynamics of gut microbiota composition following a transplant shows that even when different (germ-free) clones are grown under similar conditions, and receive a same bacterial inoculum, they can build different - genotype-dependent - gut microbial communities (see methods lines 675-752 and results lines 200-249 + Figure 5). While the inoculum effect was dominant in early life stages (soon after the transplant), the gut microbiota composition rapidly started to diverge between recipient genotypes (already after 1 week). Hence, if different genotypes harbor different microbial communities, it is likely not due to maternally transmitted bacteria, but rather to internal sorting processes (e.g. immunity). But we agree with Prof. Hairston that it would indeed be interesting to test the relative effects of genotype and gut microbiota in populations that have experienced natural selection in good and poor food environments (we mention it lines 400-403 of the revised version of the manuscript). It would also be interesting to make these clones germ-free and expose them to a same bacterial inoculum, to see if the structure of their gut microbiota would differ.

The authors may want to cite the excellent study by Jiang et al. (2015. Microgeographic adaptation to toxic cyanobacteria in two aquatic grazers. Limnology and Oceanography 60:947-956) which explores local adaptation in Daphnia populations from lakes with high and low cyanobacterial densities. I would have said that this study also showed genetic adaptation in the grazers, but because those investigators collected their animals directly from the water column and only reared them in the lab through three generations before testing for response to diet, it is possible, in light of the present study, that the differences observed were at least in part the product of differences in maternally transmitted gut microbiota. However, even in this case, it would be necessary to address the fact that all clones in that study were reared for those three generations on only good food before being experimentally exposed of good and poor food treatments of a single generation.

We now cite the study of Jiang et al. (2015) (lines 265-266). We agree with Prof. Hairston, and would do not claim that there is no genetic effect on *Daphnia* tolerance to cyanobacteria. The point we wanted to make is that in many circumstances the microbiota might actually play a key role.

Two more minor questions whose answers would strengthen the paper are: 1. Do the authors know that the strain of Microcystis aeruginosa they used was actually toxic? Many strains of this species have high concentrations of the toxin microcystin, but other strains, readily obtained from culture collections, lack the toxin. We need to know which it is for this study, especially since the authors appear to assume in the discussion that the strain is toxic and not just nutritionally poor quality (i.e., lacking essential fatty acids).

We used the strain PCC 7806 (Wild type), which is known to produce microcystin (see e.g. Hautala et al. 2012, or Rohrlack et al. 2001). We recently performed experiments in the laboratory where we successfully extracted microcystins from our *Microcystis* culture (measures were performed through both HPLC and Elisa test, showing high concentrations of microcystins). The name of the strain we used is now specified in the methods (lines 107 and 472-473).

And 2. Did the authors do a metagenomic scan of the guts of the "germ-free" Daphnia to see what if any microbiota were present before feeding started? Perhaps the latter wasn't possible since, I suppose, it would have to have been carried out on juveniles. Still it would be very interesting to see how the microbiota composition developed over time.

We did not perform microbiome analysis in germ-free animals, but the presence of bacteria in germ-free *Daphnia* has been investigated previously in the lab (through qPCR), and confirmed the absence of any bacteria after disinfection (M. Callens, PhD thesis, 2017). In this revised version of our manuscript, we performed a new transplant experiment, in which we followed the dynamics of the gut microbiota composition over time (1, 7 and 14 days after the transplant; see methods lines 675-752 and results lines 200-249 + Figure 5). Gut dissection was possible on small juveniles.

Finally, there are other questions worth considering for this system. What benefit do the microbiota get from making the *Daphnia* less sensitive to cyanobacteria in their diet? Do these microbes benefit directly because they are also making the cyanobacteria more usable for themselves? Does the *Daphnia* make the cyanobacteria more available to the microbes? Does a healthy *Daphnia* make life better for its microbes in some other way? Is there a cost to the microbes for making the cyanobacteria more usable so that in the absence of cyanobacteria a different group of microbes have an advantage? I don't suggest that the authors should have answers to such questions, but it would be good for them to indicate to their readers that they recognize that such questions are critical to answer at some point.

We agree with Prof. Hairston that to understand the evolutionary dynamics of host-microbiota associations in *Daphnia*, it would be important to determine how far the symbionts draw benefits from this relationship. Collecting and freezing microbiota at different points during a host-symbiont coevolutionary experiment, together with reciprocal microbiota inoculations, could help assess local adaptation of symbionts to their host following the general methodology of Red Queen studies. We think, however, that such questions are out the scope of the present study. We do refer to the conceptual question briefly in lines 413-418 of the revised version of the manuscript.

Very specific, small editing comments.

Line(s)

18 "... threats to livestock..." Performed accordingly.

49 Nowhere in the main body of the paper are we told what species of *Daphnia* is used. That information, currently only found in the Methods, should be mentioned up front. It is now mentioned that we used *Daphnia magna*.

97 "... genes that respond to ..." This sentence has been removed.

150 "... our study improves our understanding..." This sentence has been removed.

152-156 I think these two sentences are getting too far from the main topic. I suggest omitting them. These sentences have been removed.

Fig. 1 I've always found smiley faces rather annoying, so their presence here seems too cute to me. But that's just my own bias. The smiley faces have been removed.

177;1 88 "... on survival following exposure to toxic..." [Also, here is an example of calling the *Microcystis* toxic without evidence having been presented.] It is now specified that we used the toxic strain PCC7806 (Wild type) (lines 107 and 472-473).

Fig. 3 put the stats results on (a) and (c) even if they are ns. All the stats are now provided in the text, and not directly on the figures.

190-191 (c) and (d) should be (b) and (c) Performed accordingly.

Fig. 4 Seems like the information in panels (a) and (b) are a bit redundant. I think (b) could be moved to the on line content. Also, the grey circles (not "spheres") in (b) are pretty difficult to see. I also don't think the rarefaction curves need to be presented here. The comparison can be given in the text and the figure put in the on line content. This figure has now moved to online content (Supplementary figure 4). Rarefaction curves were removed, as well as the panel a.

401 "genotypes" Has been modified.

406 "... (each containing 60 ..." Performed accordingly.

406-407 "... jar was fed a green..." Performed accordingly.

412 "...on the condition of the ..." Performed accordingly.

415 "... kept the same among all jars." Performed accordingly.

420 "... were subjected to these diets for six months." Performed accordingly.

425 "... deionized sterile water..." Performed accordingly.

452 "For this purpose..." Performed accordingly.

558 "... were chosen as ..." Performed accordingly.

Response to Reviewer #3:

In this study, the authors investigated host protective effects of gut microbiota against toxic cyanobacteria, which is a timely and exciting topic. By using Daphnia as an experimental model, they report that resistance to toxic cyanobacteria in host is driven by the gut microbiota. Recipients that were inoculated with gut extracts from resistant donor genotypes, represent increased resistance to toxic cyanobacteria. They also provide evidence that a pre-exposure of donors to toxic cyanobacteria for 6 months resulted in enhanced protective effect of the microbiota against feeding toxic cyanobacteria (or cyanobacterial toxins). Finally, they report evidence that host genotype and diet interact to shape both structure and functionality of microbiota. Based on metagenomics analysis of gut microbiota from dissected guts of donors, they represent different bacterial communities in response to host genotypes (susceptible and resistance to toxic cyanobacteria) and diets (non-toxic green algae and toxic cyanobacteria). While this study sounds initially appealing, I find it unsatisfactory and preliminary since several major conclusions are partly supported by the reported observations and several important controls in experiments are missing, often leading important conclusions that are likely not true.

We thank the reviewer for his/her assessment. We respectfully disagree with his/her suggestion that the study is preliminary and that important controls are missing. We think our results provide strong evidence for the conclusions we draw. In the revised version of the manuscript, we try to be as clear as possible in formulating our conclusions and what we can / cannot derive from our results.

Please find below a list of major issues and claims made in the manuscript that I think, are not supported by the current data, and would definitively be needed to either be strengthened or fully re-addressed for the story to be conclusive.

1. *Daphnia* genotype affects the level of resistance to toxic cyanobacteria by modulation of their gut microbiota:

In animals, transgenerational effects contain not only the vertical transmission of microbiota and epigenetic modulations which are extensively discussed in this manuscript, but also genetic variation (e.g., SNP) and immune priming. I am not insisting that the authors should evaluate SNP or immune priming of host animal. However, the authors should at least provide how the levels of systemic and local immune responses in the host Daphnia differ in response to feeding cyanobacteria. Immune tolerance as well as immune resistance is important response against invading pathogens. Thus, the authors must verify a disparity in immune responses against toxic cyanobacteria between: (i) susceptible and resistance donor genotypes; (ii) donor pre-exposed to a non-toxic green algae diet and donor pre-exposed to a toxic cyanobacteria diet; (iii) donor and recipient. This experiment will enable us to better understand not only the cyanobacteria-mediated host immune modulation that may contribute to shape the resistant phenotype of gut microbiota against co-existing pathogens, but also the strategy in Daphnia's innate immunity (immune tolerance or resistance) against toxic cyanobacteria.

We agree with reviewer 3 that immunity likely plays an important role in *Daphnia* response to cyanobacteria. A first reason for this is that the host's immunity (e.g. antimicrobial peptides) was shown to shape the gut microbiota structure in a diverse array of species. Hence, we expect that genetic variation in *Daphnia* gut microbiota structure to partly reflect genetic variation in the immune response. The immune response may also change in response to cyanobacteria, not only to fight directly these toxic organisms and their products, but also to re-shape the gut microbiota. These issues are now discussed in the revised version of the manuscript (lines 387-399 and 410-413).

While we fully agree with the reviewer that the involvement of immunity is highly interesting and deserves further study, our study has a different focus (i.e. investigating to which extent tolerance to cyanobacteria is mediated by the gut microbiota) and we think disentangling how the host immune response shapes the gut microbiota is an intensive research project by itself and is out the scope of the present paper. It indeed represents an entire research project by itself, which we (especially EM) will develop in the coming three years. We recently obtained funding for it. We also highlight these issues in a recently published review paper in *Oikos*, in which we stressed the importance of host immunity in host-microbiota interactions (Macke et al. 2017).

2. Microbiota from resistant genotypes conferred a higher resistance to recipient Daphnia than microbiota from susceptible genotype:

First, this reviewer could not find any evidence for transmission of the microbiota from donors to recipients. It would be so much helpful if the authors provide metagenomics data of recipient gut microbiota. In addition to culture-independent data (e.g., metagenomic analysis), they must provide evidence for successful colonization and/or temporal existence of the transplanted microbiota in the gut of germ-free recipients (i.e., culture-dependent experiments such as CFU test).

In the revised version of our manuscript, we performed a new transplant experiment, intended to show to what extent the gut microbiota are transmitted and how the influence of donor versus recipient genotype on gut microbiota composition changes through time. We inoculated two different recipient genotypes with different microbial inocula derived from the gut microbiota of two donor genotypes, and then monitored the gut microbiota composition of recipient *Daphnia* over time (1, 7 and 14 days after the transplant), through next generation sequencing on 16S rRNA (see methods lines 675-752 and results lines 200-249 + Figure 5). The results of this new experiment (Experiment 3 of the revised version of the manuscript) showed that (1) transplanted microbiota successfully and stably colonize the recipients, (2) the donor effect is very strong early after inoculation but dissipates, although its influence is persistent throughout the studied period of 14 days, (3) the relative importance of the recipient genotype in explaining the gut microbiota composition increases with time (reflecting a host genotype effect).

While we did not perform counts (CFU tests), in earlier experiments of our team it was also shown that inocula successfully establish in the *Daphnia* gut after a transplant (Callens et al 2016, ISME Journal).

Most importantly, to fully support the conclusion, they must reveal the candidate microbe(s) mostly responsible for the resistance against toxic cyanobacteria at the species level. According to Figure 4 and Extended data Figure 3, it is expected that microbes in the gut of Daphnia are mostly aerobic. Thus, this reviewer strongly recommends the authors isolate bacterial species (e.g., Pseudomonas sp. found from the resistant donor, as in the Extended data Figure 3) from the gut of Daphnia, and then evaluate the resistance of the gnotobiotic animals harboring the candidate bacterial species to toxic cyanobacteria. Identifying the candidate bacterial species is particularly important because Daphnia might reflect a high spatiotemporal variation of bacterial community in their gut. For example, the structure and composition of Daphnia gut microbiota differed among different donor genotypes (Figure 3), and between this study and previous study reported by Decaestecker's lab. (Callens M et al., 2015 Food availability affects the strength of mutualistic host-microbiota interactions in Daphnia magna. ISME J), suggesting absence of core microbiota in the gut of Daphnia. Therefore, the different microbial community observed in cyanobacteria pre-exposed group possibly merely reflects spatiotemporal changes in bacterial communities. Considering these issues, claiming additional experiments such as bacterial isolation and mono-association tests are not an unreasonable demand.

We agree with the reviewer that it would be very interesting to identify specific bacteria responsible for cyanobacterial resistance in *Daphnia*. However, the innovative contribution of our study is that we show, through gut microbiota transplant experiments among genotypes that differ in tolerance to toxic cyanobacteria, that resistance to cyanobacteria is mediated by gut microbiota, and that these differences in gut microbiota are in part determined by

differences in host genotype. We think it is out the scope of the present paper to determine which bacteria are responsible and through what mechanism. We do see this as an important avenue for future research, and mention this now in the revised version of the manuscript (lines 403-410).

Identifying which bacteria confer tolerance to cyanobacteria will be an intensive research program by itself. Our metagenetic analyses provide some hints. First, in the new fully replicated metagenetic analyses reported on in the revised version of the manuscript we observe a dominance of Flavobacteria in the guts of tolerant *Daphnia*. While there is no evidence that Flavobacteria can degrade cyanobacteria toxins, they are often reported to coexist with blooms of cyanobacteria and are therefore considered tolerant. Yet, our metagenetics data also show that responses are very variable, as in the pilot experiment (reported upon in the suppl information of the revised version of the manuscript) we did observe the presence of taxa that are known to degrade cyanobacteria toxins in the gut microbiota of tolerant *Daphnia*. That the gut microbiota in the two experiments differed is not surprising as the animals were exposed to pond water, the composition of which is likely variable. We can from our data not conclude whether tolerance to toxins of cyanobacteria was mediated by different bacteria taxa in the two experiments or whether the taxa degrading cyanobacteria were so low in abundance to be detected by our metagenetic analysis (being overwhelmed by the dominance of Flavobacteria) in the second, full analysis. But the results do indicate that responses may be variable. This is discussed lines 325-353.

We agree with the reviewer that the microbial community in *Daphnia* is variable, and that its composition can differ among genotypes, diets, and in response to diverse factors. This is amongst others also reflected by the differences between the pilot and full experiment discussed in the previous paragraph. Our transplant experiment investigating gut microbiota dynamics over time (Experiment 3; methods lines 675-752 and results lines 200-249 + Figure 5) shows that the microbiota structure strongly depends on exogenous microbial exposure early in life (just after the transplant), but is progressively reshaped over time, revealing a divergence between genotypes. This result shows that although the genotype contributes to shaping the gut microbiota, the pool of microbes available in the surrounding environment is also an important determinant of the microbiota assembly. Hence, indeed, *Daphnia* are expected to exhibit high spatiotemporal variation of bacterial community in their gut. This is now discussed lines 375-378. However, the results of Experiment 3 show that our observations that tolerance to cyanobacteria is mediated through gut microbiota does not simply reflect spatiotemporal variation in gut microbiota composition, but also has a host genotype effect.

3. The second paragraph in main text reporting that gut microbiota rather than *Daphnia* genotype itself drives resistance to toxic cyanobacteria is confusing. Germ-free control should be included in Figure 2b, c and d, and Figure 3.

We agree with the reviewer that claiming that *Daphnia* genotype does not have any direct effect on resistance to cyanobacteria is too excessive. We now removed such sentences from the manuscript. Our transplant experiment revealed that when all recipient genotypes are inoculated with a same microbial inoculum, all genotypes exhibited similar survival upon cyanobacterial exposure, and this survival differed depending on the donor gut microbiota. Hence, differences in fitness between tolerant and susceptible genotypes can no longer be observed when they have been transplanted with the same microbial inoculum. Conversely, strong differences were observed between recipient *Daphnia* that received different microbial inocula (i.e. significant donor effect). Indeed, survival in recipients was higher when inoculated with the microbiota of the tolerant genotypes. The fact that cyanobacterial tolerance in recipient *Daphnia* depends on the donor, and not on the recipient genotype, points to the gut microbiota as an important mediator of cyanobacterial tolerance. Our results suggest that interclonal variation in cyanobacterial tolerance is to an important degree determined by host genotype mediated differences in the gut microbiota composition. In the revised version of the manuscript, we now reformulate this more carefully, as we do not want to claim that there no direct *Daphnia* genotype effects on cyanobacteria tolerance, even though we did not observe such effects (see lines 261-276).

We respectfully disagree with the reviewer that germ-free controls are necessary for a correct interpretation of our results. We aimed to investigate to what extent differences in tolerance is mediated by differences in gut microbiota and differences in host genotype, so transplanting gut microbiota factorially across genotypes is key to the design.

The use of germ-free controls would not add much, as germ-free animals typically perform very poorly even under control conditions (e.g. Sison-Mangus et al. 2014, Callens et al. 2016). *Daphnia* need their gut microbiota. It was not our aim to show how important the presence of gut microbiota is for cyanobacteria tolerance, but rather how important (host genotype mediated) differences in gut microbiota are.

4. In Figure 4, I see only a single biological replicate for the metagenomics analysis. More biological replicates (at least $n=3$) need to be implemented for the sake of the statistical analysis robustness. Also, the number of PCR cycle (35 cycles) is not low. Add technical replicate (at least $n=3$) to minimize a potential PCR bias.

This metagenetics analysis was performed again, and we now have 3 biological replicates per treatment (see methods lines 610-673 and results lines 165-198 + Figures 4 and Supplementary Figure 3). In addition, 3 technical replicates were added for the library preparation (see methods line 636).

5. In Figure 4d, *Daphnia* pre-exposed to a toxic cyanobacteria diet represent higher values of microbial richness and evenness than *Daphnia* pre-exposed to a non-toxic green algae diet. Provide reasonable discussion about potential factor(s) influencing the microbial diversity in the gut of *Daphnia*.

The factors that may influence the microbial diversity in the gut of *Daphnia* are now briefly discussed in the revised version of the manuscript (lines 389-399).

Specific comments:

- Line 87. Correct spacing error. This part of the text has changed.

- Figure 3 legends. Correct the order. It should be a, b and c. Performed accordingly.

REVIEWERS' COMMENTS:

Reviewer #1 (Remarks to the Author):

The investigation of gut microbiological communities and their effects on fitness, physiology and protection against pathogens of the respective host is currently a hot topic in environmental medicine and ecology. Here the authors address the question in how far different gut communities determine the tolerance of the host *Daphnia* to a toxic food item, which is a bacterium itself (cyanobacterium). Four genotypes of *Daphnia*, that differ in their tolerance of a cyanobacterium, were grown for several months in the absence/presence of this cyanobacterium with regular additions of natural surface water in order to provide a somewhat natural bacterial community. Then these animals ('donors') were used for the preparation of gut homogenates that were subsequently transferred to three of these genotypes (recipient genotypes). Donors with a high tolerance resulted in elevated tolerance of the receiver, and this effect was stronger when donors had been fed cyanobacteria; sensitive donors lead to less tolerant recipient genotypes. The composition of the gut microbiota differed among tolerant and sensitive donor strains. In an additional experiment the microbiota of two recipient *Daphnia* genotypes were monitored, which revealed that both, host genotype and initial bacterial inoculum, contribute to the gut microbiome, with the effect of the host genotype increasing over time.

The paper is one of the first that links changes in microbiomes to aspects of fitness. It reports impressively strong effects of microbiota, which are new with respect to interactions of *Daphnia* and cyanobacteria, and as such it has the potential to remarkably influence our understanding of acclimation and perhaps adaptation to fluctuating and changing environments in general.

In this revised form the experimental design with replicated lines of preconditioning and subsequent replicated transplantations is fully adequate to assess the effects of transplants on fitness and to convincingly show a major role of the microbiome in mediating the tolerance of *Daphnia* to toxic cyanobacteria.

The authors have adequately responded to all major points that I had addressed in response to an earlier version of this ms. In particular, another fully replicated transplantation experiment has been performed, in which the microbiome of recipient *Daphnia* genotypes has been analysed over time.

I have few minor points that are listed below.

Title: change into 'Host-genotype dependent gut...'

line 20: 'genetic and plastic variation in the gut microbiota'? I do not know what is meant by 'plastic variation'. I assume it means physiological plasticity of gut microbiota? If so, please see my comment to lines 293-296, where I basically argue that physiological plasticity is not the only possible explanation. In conclusion, I feel that you have not demonstrated physiological plasticity in gut microbiota - it is one of at least two possible interpretations of your results. Therefore, please remove 'plastic' from this sentence.

line 19-21: Please rephrase to: "we here show that in the freshwater crustacean *Daphnia magna*, in addition to host genotype, genetic variation in the gut microbiota mediates tolerance to toxic cyanobacteria, ...". The current version is not adequately communicating the effects of the host genome, which was a major point of concern in all three reviews.

line 95: see above my comments to line 20.

line 112: replace 'R1 and R2' by 'T1 and T2'

line 123: 'subsequently exposed'? This reads as if, after exposure to pondwater, they then were exposed to the different diets. What you probably mean is that they were exposed to the two different food treatments in the pondwater? Please rephrase accordingly.

line 293-296: these lines need a bit more of clarification. ..Please add 'of the donor strain' after 'upon prior exposure to cyanobacteria'. To me the conclusion that these two findings suggest physiological acclimation of given bacterial taxa is not the only possible interpretation: The observed enhanced tolerance upon prior exposure of *Daphnia* donor strains to cyanobacteria is based on (transplantation of) microbiota that resulted from a period of 12 generations of exposure

to the toxic cyanobacterium. The effects of cyanobacterial exposure on taxonomic composition of the microbiota were monitored over 14 days. Therefore, another possible explanation is that 14 days were too short to detect effects of cyanobacterial exposure on taxonomic diversity of the microbiota. In the light of this, the fairly speculative lines 301-304 might be removed or the alternative explanation should be included.

line 307. Has it really been experimentally shown that mucilage reduces assimilation of cyanobacterial carbon? If this is an evidence-based and not an experiment based statement, I suggest to remove this statement / reference.

line 405: If degradation of microcystin by *Daphnia* (and associated gut bacteria) would occur, this would result in an overall reduction of microcystins. However, Sadler and Von Elert (2014) did not find evidence for degradation of microcystins when the same cyanobacterial strain as was used here (PCC 7806) was ingested and digested by *D. magna*. Therefore, it should be mentioned here that experimental evidence does not support the idea of a significant degradation of microcystin during *Daphnia* digestion (Sadler and Von Elert 2014).

line 424: what is 'plastic variation'? Do you mean physiological plasticity of bacterial taxa - please see my comment above to lines 293-296. May be rephrase to 'host genotype mediated and environmentally mediated variation in gut microbiota structure'...

line 429-434: Reviewer 2 has made the point that the results of (Airston et al. 1999) are strong evidence for an important role of host genotype. I would like the authors to please refer to this widely known seminal paper here and shortly discuss that the identical genetic bacterial pool during the culturing of those clones may provide a strong argument in favor of the importance of the host genotype, whereas, in the light of the results presented here, one might speculate that the different *Daphnia* genotypes had acquired different microbiota which then lead to the differences in tolerance.

line 966-974: Legend of Fig 4. Add the information that these are the donor-populations.

line 977: Legend of Fig 5. Replace 'PCoA' by Principal Component Analysis

Eric von Elert

Reference List

Hairston, N. G., W. Lampert, C. E. Cáceres, C. L. Holtmeier, L. J. Weider, U. Gaedke, J. M. Fischer, J. A. Fox, and D. M. Post. 1999. Rapid evolution revealed by dormant eggs. *Nature* 401: 446.
Sadler, T., and E. Von Elert. 2014. Dietary exposure of *Daphnia* to microcystins: No in vivo relevance of biotransformation. *Aquatic Toxicology* 150: 73-82.

Reviewer #3 (Remarks to the Author):

The major comments I raised in the first revision was to improve the quality of current manuscript, but NOT to recommend the authors for considering the several critical points in further separated study. I still find that the several major conclusions the authors insist are not sufficiently supported by current data.

Original comment

1. *Daphnia* genotype affects the level of resistance to toxic cyanobacteria by modulation of their gut microbiota:

In animals, transgenerational effects contain not only the vertical transmission of microbiota and epigenetic modulations which are extensively discussed in this manuscript, but also genetic variation (e.g., SNP) and immune priming. I am not insisting that the authors should evaluate SNP

or immune priming of host animal. However, the authors at least provide how the levels of systemic and local immune responses in host Daphnia differ in response to feeding cyanobacteria. Immune tolerance as well as immune resistance is important response against invading pathogens. Thus, the authors must verify a disparity in immune responses against toxic cyanobacteria between: (i) susceptible and resistance donor genotypes; (ii) donor pre-exposed to a non-toxic green algae diet and donor pre-exposed to a toxic cyanobacteria diet; (iii) donor and recipient. This experiment will enable us to better understand not only the cyanobacteria-mediated host immune modulation that may contribute to shape the resistant phenotype of gut microbiota against co-existing pathogens, but also the strategy in Daphnia's innate immunity (immune tolerance or resistance) against toxic cyanobacteria.

Author response

While we fully agree with the reviewer that the involvement of immunity is highly interesting and deserves further study, our study has a different focus (i.e. investigating to which extent tolerance to cyanobacteria is mediated by the gut microbiota) and we think disentangling how the host immune response shapes the gut microbiota is an intensive research project by itself and is out the scope of the present paper. It indeed represents an entire research project by itself, which we (especially EM) will develop in the coming three years. We recently obtained funding for it. We also highlight these issues in a recently published review paper in *Oikos*, in which we stressed the importance of host immunity in host-microbiota interactions (Macke et al. 2017).

New comment

I recommended that the authors at least provide how the levels of systemic and local immune responses in host Daphnia differ in response to feeding cyanobacteria, but NOT to disentangle how the host immune response shapes the gut microbiota. I believe that conducting qRT-PCR to target the several AMP encoding genes not represent an entire research project by itself. I insist that the authors must verify a disparity in immune responses against toxic cyanobacteria between: (i) susceptible and resistance donor genotypes; (ii) donor pre-exposed to a non-toxic green algae diet and donor pre-exposed to a toxic cyanobacteria diet; (iii) donor and recipient.

Original comment

3. The second paragraph in main text reporting that gut microbiota rather than Daphnia genotype itself drives resistance to toxic cyanobacteria is confusing. Germ-free control should be included in Figure 2b, c and d, and Figure 3.

Author response

We respectfully disagree with the reviewer that germ-free controls are necessary for a correct interpretation of our results. We aimed to investigate to what extent differences in tolerance is mediated by differences in gut microbiota and differences in host genotype, so transplanting gut microbiota factorially across genotypes is key to the design. The use of germ-free controls would not add much, as germ-free animals typically perform very poorly even under control conditions (e.g. Sison-Mangus et al. 2014, Callens et al. 2016). Daphnia need their gut microbiota. It was not our aim to show how important the presence of gut microbiota is for cyanobacteria tolerance, but rather how important (host genotype mediated) differences in gut microbiota are.

New comment

The original comment 3 was relevant to the comment 1 described above. The core conclusion of the entire manuscript is that "gut microbiota drives resistance to toxic cyanobacteria". Accordingly, the authors must provide a simple evidence that the gut microbiota rather than other host genomic, epigenetic and/or immunogenic factors is responsible for resistance to toxic cyanobacteria. To this end, I believe that using germ-free animal is the easiest way. If generation of the germ-free animal is not applicable, please use several combination of antibiotics to disturb the original resistance microbial phenotype of donor host, and then observe the recipient phenotype to toxic cyanobacteria after microbial transplantation.

REVIEWERS' COMMENTS:

Reviewer #1 (Remarks to the Author):

The investigation of gut microbiological communities and their effects on fitness, physiology and protection against pathogens of the respective host is currently a hot topic in environmental medicine and ecology. Here the authors address the question in how far different gut communities determine the tolerance of the host *Daphnia* to a toxic food item, which is a bacterium itself (cyanobacterium). Four genotypes of *Daphnia*, that differ in their tolerance of a cyanobacterium, were grown for several months in the absence/presence of this cyanobacterium with regular additions of natural surface water in order to provide a somewhat natural bacterial community. Then these animals ('donors') were used for the preparation of gut homogenates that were subsequently transferred to three of these genotypes (recipient genotypes). Donors with a high tolerance resulted in elevated tolerance of the receiver, and this effect was stronger when donors had been fed cyanobacteria; sensitive donors lead to less tolerant recipient genotypes. The composition of the gut microbiota differed among tolerant and sensitive donor strains. In an additional experiment the microbiota of two recipient *Daphnia* genotypes were monitored, which revealed that both, host genotype and initial bacterial inoculum, contribute to the gut microbiome, with the effect of the host genotype increasing over time.

The paper is one of the first that links changes in microbiomes to aspects of fitness. It reports impressively strong effects of microbiota, which are new with respect to interactions of *Daphnia* and cyanobacteria, and as such it has the potential to remarkably influence our understanding of acclimation and perhaps adaptation to fluctuating and changing environments in general. In this revised form the experimental design with replicated lines of preconditioning and subsequent replicated transplantations is fully adequate to assess the effects of transplants on fitness and to convincingly show a major role of the microbiome in mediating the tolerance of *Daphnia* to toxic cyanobacteria.

The authors have adequately responded to all major points that I had addressed in response to an earlier version of this ms. In particular, another fully replicated transplantation experiment has been performed, in which the microbiome of recipient *Daphnia* genotypes has been analysed over time.

I have few minor points that are listed below.

We thank Eric von Elert for his positive and constructive comments. Please find our responses below.

- Title: change into 'Host-genotype dependent gut...'

Performed accordingly

- line 20: 'genetic and plastic variation in the gut microbiota'? I do not know what is meant by 'plastic variation'. I assume it means physiological plasticity of gut microbiota? If so, please see my comment to lines 293-296, where I basically argue that physiological plasticity is not the only possible explanation. In conclusion, I feel that you have not demonstrated physiological plasticity in gut microbiota - it is one of at least two possible interpretations of your results. Therefore, please remove 'plastic' from this sentence.

By “plastic variation in the gut microbiota”, we in fact meant “environmentally induced” variation (in terms of both taxonomic composition and functionality), in opposition to the variation of gut microbiota explained host genotype (i.e. “genetic variation”). The microbiota was considered like an extended host phenotype, shaped by both host genetics and the environment. This is what we meant by “genetic” and “plastic” variations.

To avoid any confusion, the terms “genetic variation” and “plastic variation” have been removed from this revised version, and replaced by “host-genotype induced variation” and “environmentally induced variation” (see line 25 of the revised manuscript).

- line 19-21: Please rephrase to: "we here show that in the freshwater crustacean *Daphnia magna*, in addition to host genotype, genetic variation in the gut microbiota mediates tolerance to toxic cyanobacteria, ...". The current version is not adequately communicating the effects of the host genome, which was a major point of concern in all three reviews.

*The whole abstract has been changed to satisfy the Editor’s requirement. The sentence is now “Here we use gut microbiota transplants to show that, in the freshwater crustacean *Daphnia magna*, **in addition to host genotype**, gut microbiota mediates tolerance to toxic cyanobacteria” (see lines 17-20 of the revised manuscript)*

- line 95: see above my comments to line 20.

The term “plastic variation” has been removed. The sentence has been reformulated in accordance with the Reviewer’s comments on lines 19-21 (see lines 104-105 of the revised manuscript).

- line 112: replace 'R1 and R2' by 'T1 and T2'

Performed accordingly (see line 122 of the revised manuscript).

- line 123: 'subsequently exposed'? This reads as if, after exposure to pondwater, they then were exposed to the different diets. What you probably mean is that they were exposed to the two different food treatments in the pondwater? Please rephrase accordingly.

**Daphnia* were indeed exposed to the two food treatments in the pond water. The word “subsequently” has thus been removed from the sentence (see line 133 of the revised manuscript).*

- line 293-296: these lines need a bit more of clarification. ..Please add 'of the donor strain' after 'upon prior exposure to cyanobacteria'. To me the conclusion that these two findings suggest

physiological acclimation of given bacterial taxa is not the only possible interpretation: The observed enhanced tolerance upon prior exposure of *Daphnia* donor strains to cyanobacteria is based on (transplantation of) microbiota that resulted from a period of 12 generations of exposure to the toxic cyanobacterium. The effects of cyanobacterial exposure on taxonomic composition of the microbiota were monitored over 14 days. Therefore, another possible explanation is that 14 days were too short to detect effects of cyanobacterial exposure on taxonomic diversity of the microbiota. In the light of this, the fairly speculative lines 301-304 might be removed or the alternative explanation should be included.

The sentence has been changed to “Given that we did observe an enhanced tolerance upon prior exposure of the donor strains to cyanobacteria, ...” (see line 309 of the revised manuscript).

*Concerning the interpretation of our results (i.e. suggesting physiological acclimation of given bacterial taxa), we think there was a misunderstanding about our experimental design. The gut microbiota of our donor populations was not characterized after 14 days of exposure to the two types of diet, but after 58 generations (see details below). Hence, we think that the interpretation suggested by Reviewer 1 (i.e. 14 days of exposure is not enough to observe an effect of the diet on the gut microbiota composition) is not suitable here. The misunderstanding may come from a confusion with Experiment 3 (see lines 709-787), in which we monitored the gut microbiota dynamics over a period of 14 days in recipient *Daphnia* inoculated with different types of microbial inocula. In experiment 3, however, we did not test for the effects of the diet.*

*The gut microbiota characterization of the donor *Daphnia* genotypes exposed to either a cyanobacterial or a green algal diet was performed on the same donor populations as those described in the Experiment 1, which were continuously maintained in the laboratory under these dietary conditions. This characterization was performed two times. The first time, it was performed on a subset of populations (i.e. preliminary test, with 1 replicate per sample and per diet; see supplementary information), approximately at the same time as the transplant experiment (i.e. after 12 generations of exposure to the two diets). In this preliminary test, we were not able to analyze the data properly, because we did not have enough replicates, but the results suggested a possible effect of the diet on the microbiota composition (see supplementary Figure 4). We thus decided to characterize again the gut microbiota of these experimental populations (still the same populations as those used in Experiment 1, which were continuously maintained in the laboratory under the same dietary conditions), but this time on the whole set of populations (3 replicates per genotype and diet). This second characterization of our donor populations, which did not reveal any effect of the diet on microbiota composition (see Figure 4), was performed 1.5 years after Experiment 1, thus after approximately 58 generations of exposure to the two types of diet.*

*To avoid any confusion, we now clearly mention in the methods that the microbiota characterization was performed on the same donor populations as those described in Experiment 1, after 58 generations of exposure to the two types of diet (see lines 645-649 of the revised manuscript). In addition, we removed the sentence “The four *Daphnia* populations were subjected to these diets for six months” from the description of Experiment 1 in the methods, because it was confusing. Instead, we say that the transplant experiment was performed “after six months (i.e. 12 generations) of exposure of the donors to the two types of diet (see lines 568-572 of the revised manuscript).*

- line 307. Has it really been experimentally shown that mucilage reduces assimilation of cyanobacterial carbon? If this is an evidence-based and not an experiment based statement, I suggest to remove this statement / reference.

This statement has been removed (see lines 319-321 of the revised manuscript)

line 405: If degradation of microcystin by Daphnia (and associated gut bacteria) would occur, this would result in an overall reduction of microcystins. However, Sadler and Von Elert (2014) did not find evidence for degradation of microcystins when the same cyanobacterial strain as was used here (PCC 7806) was ingested and digested by *D. magna*. Therefore, it should be mentioned here that experimental evidence does not support the idea of a significant degradation of microcystin during Daphnia digestion (Sadler and Von Elert 2014).

We now mention that the study of Sadler and von Elert did not reveal any biodegradation of microcystins during the digestion of Microcystis cells in Daphnia (see lines 422-423 of the revised manuscript).

Based on our data, we however think that experimental conditions (i.e. the microbial community available to the Daphnia at the moment of the experiment, or the Daphnia genotype) may influence the results of such studies. In Sadler and von Elert (2014), the absence of microcystin degradation may also be due to (1) a too short exposure of Daphnia to Microcystis (i.e. 24h were maybe not enough to induce microcystin degradation by the bacteria present in the Daphnia gut) and (2) a gut microbial community not diverse enough to respond to the Microcystis exposure (i.e. experimental Daphnia were subjected to several rinsing steps following exposure to cyanobacteria, which is expected to disturb the gut microbial community). We thus added this sentence: "The occurrence of microcystin degradation may, however, depend on a combination of factors, including e.g. the bacterial community available to the Daphnia, the Daphnia genotype or the duration of Daphnia exposure to the cyanotoxins" (see lines 423-425 of the revised manuscript).

We would also like to mention to Referee 1 that we recently performed a preliminary in-vitro experiment, in which we monitored the degradation of microcystins by the Daphnia gut microbiota, using HPLC. Cyanobacterial toxins were first extracted from Microcystis cells using sonication (until Microcystis cells were not observable anymore under the microscope). The obtained product was subsequently divided into control jars, and jars containing Daphnia gut microbiota extracts. Microcystin concentration in the different jars was monitored over time using HPLC. The results, although very preliminary, showed a decrease in microcystin concentration in jars containing the gut microbiota, but not in the control, suggesting a microcystin-degrading activity in the Daphnia gut microbiota.

line 424: what is 'plastic variation'? Do you mean physiological plasticity of bacterial taxa - please see my comment above to lines 293-296. May be rephrase to 'host genotype mediated and environmentally mediated variation in gut microbiota structure'...

The sentence has been rephrased to “both host genotype mediated and environmentally mediated variation in gut microbiota structure affect host fitness and its response to environmental pressure, here exposure to toxic cyanobacteria” (see lines 445-448 of the revised manuscript).

line 429-434: Reviewer 2 has made the point that the results of Hairston et al. 1999 are strong evidence for an important role of host genotype. I would like the authors to please refer to this widely known seminal paper here and shortly discuss that the identical genetic bacterial pool during the culturing of those clones may provide a strong argument in favor of the importance of the host genotype, whereas, in the light of the results presented here, one might speculate that the different Daphnia genotypes had acquired different microbiota which then lead to the differences in tolerance.

We now refer to this seminal paper and discuss the strong argument it provides in favor of the importance of host genotype. We added this paragraph: “In a seminal resurrection ecology study, Hairston et al. found strong interclonal variation in Daphnia tolerance to cyanobacteria, and convincingly demonstrated that tolerance of Daphnia genotypes to cyanobacteria increased over time in response to increasing eutrophication. In this study, Daphnia genotypes were hatched from resting eggs, and exposed to identical microbial pool throughout the experiment, providing a strong argument in favor of the importance of host genotype. In light of our results, one might speculate that these genotypes had acquired different gut microbiota, e.g. through selective recruitment mechanisms, which then lead to differences in tolerance.” (see lines 452-459 of the revised manuscript).

line 966-974: Legend of Fig 4. Add the information that these are the donor-populations.

Performed accordingly (see line 1029 of the revised manuscript).

line 977: Legend of Fig 5. Replace 'PCoA' by Principal Component Analysis

Performed accordingly (see lines 1046 and 1060 of the revised manuscript).

Eric von Elert

Reference List

Hairston, N. G., W. Lampert, C. E. Cáceres, C. L. Holtmeier, L. J. Weider, U. Gaedke, J. M. Fischer, J. A. Fox, and D. M. Post. 1999. Rapid evolution revealed by dormant eggs. *Nature* 401: 446.

Sadler, T., and E. Von Elert. 2014. Dietary exposure of Daphnia to microcystins: No in vivo relevance of biotransformation. *Aquatic Toxicology* 150: 73-82.

Reviewer #3 (Remarks to the Author):

The major comments I raised in the first revision was to improve the quality of current manuscript, but NOT to recommend the authors for considering the several critical points in further separated study. I still find that the several major conclusions the authors insist are not sufficiently supported by current data.

We thank Reviewer 3 for his comments. Based on the recommendation of the Editor, we respectfully decided not to perform the additional experiments required by Reviewer 3. Please find our detailed response below.

1. Original comment

Daphnia genotype affects the level of resistance to toxic cyanobacteria by modulation of their gut microbiota: In animals, transgenerational effects contain not only the vertical transmission of microbiota and epigenetic modulations which are extensively discussed in this manuscript, but also genetic variation (e.g., SNP) and immune priming. I am not insisting that the authors should evaluate SNP or immune priming of host animal. However, the authors at least provide how the levels of systemic and local immune responses in host Daphnia differ in response to feeding cyanobacteria. Immune tolerance as well as immune resistance is important response against invading pathogens. Thus, the authors must verify a disparity in immune responses against toxic cyanobacteria between: (i) susceptible and resistance donor genotypes; (ii) donor pre-exposed to a non-toxic green algae diet and donor pre-exposed to a toxic cyanobacteria diet; (iii) donor and recipient. This experiment will enable us to better understand not only the cyanobacteria-mediated host immune modulation that may contribute to shape the resistant phenotype of gut microbiota against co-existing pathogens, but also the strategy in Daphnia's innate immunity (immune tolerance or resistance) against toxic cyanobacteria.

Author response

While we fully agree with the reviewer that the involvement of immunity is highly interesting and deserves further study, our study has a different focus (i.e. investigating to which extent tolerance to cyanobacteria is mediated by the gut microbiota) and we think disentangling how the host immune response shapes the gut microbiota is an intensive research project by itself and is out the scope of the present paper. It indeed represents an entire research project by itself, which we (especially EM) will develop in the coming three years. We recently obtained funding for it. We also highlight these issues in a recently published review paper in *Oikos*, in which we stressed the importance of host immunity in host-microbiota interactions (Macke et al. 2017).

New comment

I recommended that the authors at least provide how the levels of systemic and local immune responses in host Daphnia differ in response to feeding cyanobacteria, but NOT to disentangle how the host immune response shapes the gut microbiota. I believe that conducting qRT-PCR to target the several AMP encoding genes not represent an entire research project by itself. I insist that the authors must verify a disparity in immune responses against toxic cyanobacteria between: (i) susceptible and resistance donor genotypes; (ii) donor pre-exposed to a non-toxic green algae diet and donor pre-exposed to a toxic cyanobacteria diet; (iii) donor and recipient.

We agree with Referee 3 that investigating Daphnia immune responses against toxic cyanobacteria will certainly provide important complementary information about Daphnia tolerance to cyanobacteria. We however still think that, although very interesting, this question is not the scope of the present study. Interactions between Daphnia and cyanobacteria may indeed involve immunity, but also other factors like epigenetics, chymotrypsins, ... that may affect the interaction. Our study does not aim at unravelling the whole, likely highly complex, interaction network between Daphnia, their gut microbiota and cyanobacteria, but rather focuses on the microbiota-mediated part of tolerance. Following the recommendation of the Editor (see above), we will thus not perform, in the context of the present study, the additional experiments required by Reviewer 3.

We mention several times in the discussion that addressing the immune aspects of Daphnia-cyanobacteria interactions will be key in future studies. Especially, we mention that “obtaining insight in the role of Daphnia immunity in structuring the gut microbiota, and comparing the immune profile of Daphnia among genotypes (e.g. tolerant versus susceptible) and diets (e.g. toxic versus non-toxic), will be key. This would enable us to better understand not only the host immune modulation that contributes to shape the tolerant phenotype of gut microbiota against cyanobacteria, but also the strategy in Daphnia's innate immunity (immune tolerance or resistance) against toxic cyanobacteria themselves” (see lines 431-434 of the revised manuscript).

3. Original comment

The second paragraph in main text reporting that gut microbiota rather than *Daphnia* genotype itself drives resistance to toxic cyanobacteria is confusing. Germ-free control should be included in Figure 2b, c and d, and Figure 3.

Author response

We respectfully disagree with the reviewer that germ-free controls are necessary for a correct interpretation of our results. We aimed to investigate to what extent differences in tolerance is mediated by differences in gut microbiota and differences in host genotype, so transplanting gut microbiota factorially across genotypes is key to the design. The use of germ-free controls would not add much, as germ-free animals typically perform very poorly even under control conditions (e.g. Sison-Mangus et al. 2014, Callens et al. 2016). *Daphnia* need their gut microbiota. It was not our aim to show how important the presence of gut microbiota is for cyanobacteria tolerance, but rather how important (host genotype mediated) differences in gut microbiota are.

New comment

The original comment 3 was relevant to the comment 1 described above. The core conclusion of the entire manuscript is that “gut microbiota drives resistance to toxic cyanobacteria”. Accordingly, the authors must provide a simple evidence that the gut microbiota rather than other host genomic, epigenetic and/or immunogenic factors is responsible for resistance to toxic cyanobacteria. To this end, I believe that using germ-free animal is the easiest way. If generation of the germ-free animal is not applicable, please use several combination of antibiotics to disturb the original resistance microbial phenotype of donor host, and then observe the recipient phenotype to toxic cyanobacteria after microbial transplantation.

Our results indicate that prior to any manipulation of gut microbiota, Daphnia genotypes strongly differed in their level of tolerance to toxic cyanobacteria (Figure 1 and Supplementary Figure 1), which is consistent with previous studies showing interclonal variation in tolerance (e.g. Hairston et al. 1999). Interclonal variation in Daphnia tolerance to cyanobacteria, however, completely disappeared when Daphnia were made germ-free and received an identical microbial inoculum. Hence, homogenizing the gut microbiota among clones was enough to get rid of interclonal variation in tolerance to cyanobacteria. We think this result is sufficient to prove that gut microbiota is an important mediator of Daphnia tolerance to cyanobacteria. The fact that tolerance in recipient mirrored tolerance in donors, and that tolerant and susceptible genotypes exhibited very different microbial communities, provided additional support to this conclusion.

In our transplant experiment, when recipient Daphnia were made germ-free, only gut microbes were removed, while other genomic, epigenetic or immunogenic factors were normally not changed (at least, not more than they would be in the experiment suggested by reviewer 3). If interclonal variation in tolerance to cyanobacteria was mediated by one of these factors, we should thus still observe differences among our recipient genotypes, which is not the case. Hence, we think the disappearance of interclonal variation in tolerance to cyanobacteria in recipient Daphnia is necessarily due to the disturbance of their gut microbial community.

Based on the arguments developed above, and on the recommendation of the Editor not to perform additional experiments, we respectfully decided not to address the request of Reviewer 3.